# Tumor cell-directed STING agonist antibody-drug conjugates induce type III interferons and anti-tumor innate immune responses

Naniye Malli Cetinbas [1] ✉, Travis Monnell[1], Jahna Soomer-James[1], Pamela Shaw [1], Kelly Lancaster[1], Kalli C. Catcott [1], Melissa Dolan[1], Rebecca Mosher[1], Caitlin Routhier [1], Chen-Ni Chin[1], Dorin Toader[1], Jeremy Duvall[1], Raghida Bukhalid[1], Timothy B. Lowinger [1] & Marc Damelin [1] ✉

Activating interferon responses with STING agonists (STINGa) is a current cancer immunotherapy strategy, and therapeutic modalities that enable tumor-targeted delivery via systemic administration could be beneficial. Here we demonstrate that tumor cell-directed STING agonist antibody-drug-conjugates (STINGa ADCs) activate STING in tumor cells and myeloid cells and induce anti-tumor innate immune responses in in vitro, in vivo (in female mice), and ex vivo tumor models. We show that the tumor cell-directed STINGa ADCs are internalized into myeloid cells by Fcγ-receptor-I in a tumor antigen-dependent manner. Systemic administration of STINGa ADCs in mice leads to STING activation in tumors, with increased anti-tumor activity and reduced serum cytokine elevations compared to a free STING agonist. Furthermore, STINGa ADCs induce type III interferons, which contribute to the anti-tumor activity by upregulating type I interferon and other key chemokines/cytokines. These findings reveal an important role for type III interferons in the anti-tumor activity elicited by STING agonism and provide rationale for the clinical development of tumor cell-directed STINGa ADCs.

The STING (STimulator of InterferoN Genes) pathway is a critical component of anti-viral and anti-tumor innate immune responses[1–3]. In normal physiology, STING is activated by its natural agonist cGAMP, which is generated by the pattern recognition receptor cGAS in response to cytosolic dsDNA[4,5]. cGAMP binding to STING activates TBK1/IKK signaling followed by IRF3- and NF-κB-dependent production of type I interferons (IFNs) and other inflammatory cytokines/chemokines[6]. STING signaling has been shown to mediate type III IFN induction in response to viral infections[7], but this pathway has not been well-studied in the context of tumors.

Considering the immunostimulatory and anti-tumor effects of type I IFN responses, including innate immune activation, increased antigen presentation, immune cell infiltration, and tumor specific CD8 + T cell activation[8–10], STING has been actively pursued as a target for cancer immunotherapy. Several small molecule STINGa have been developed

and have exhibited anti-tumor immune activity in preclinical models, and various intratumorally (IT) or systemically administered STINGa are currently in clinical development[11]. The recent data from a phase 1 trial of an IT-administered STINGa demonstrated significant shrinkage in the injected tumors but no changes in the distal lesions, such that the overall anti-tumor activity was minimal[12]. While these findings suggest that STING agonism could confer clinical benefit, they also highlight the importance of tumor accessibility via systemic delivery of the STINGa. However, the systemic administration of free STINGa has toxicity concerns due to undesired STING activation in peripheral cells[11]. The success of small molecule STINGa could be further limited by the growing evidence that STING pathway activation can be immune-suppressive in certain cell types[13]; several studies in preclinical models demonstrated the negative impact of STING pathway activation on T cell and B cell viability and fitness[14–17]. Therefore, it may be advantageous to achieve

[1]Mersana Therapeutics Inc. Cambridge MA, Cambridge, USA. ✉e-mail: ncetinbas@mersana.com; mdamelin@mersana.com

targeted delivery of STINGa to specific cell types within the tumor microenvironment.

Type I IFN responses in antigen presenting cells such as dendritic cells and macrophages[8,10] as well as stromal cells[18,19] have been shown to mediate the anti-tumor activity of STINGa. Although some studies reported that the cancer cells are unresponsive to STING agonism due to epigenetic silencing of the *STING* gene[20,21] or suppression of STING signaling[22,23], others indicated that cancer cell STING is required for anti-tumor immune responses induced by radiation therapy and DNA-damaging reagents in preclinical tumor models[24,25]. Moreover, cancer cell STING expression and perinuclear localization correlate with better prognosis and response to immuno-therapy in clinical settings[26,27]. Thus, growing evidence supports the notion of productive STING signaling in myeloid cells and cancer cells, however the anti-tumor effects of STING activation in these cell types via a targeted STINGa delivery approach, such as an antibody-drug conjugate (ADC) has not been studied.

The ADC is a clinically validated therapeutic modality in which a drug "payload" is conjugated to an antibody, allowing tumor-targeted drug delivery with systemic administration. ADCs deliver payload to the target (antigen-expressing) cells, typically tumor cells, via antigen-binding and internalization, usually by endocytosis[28]. In addition, Fc-mediated interactions of antigen-bound antibodies with Fcγ-receptors (FcγRs) on myeloid cells can lead to FcγR clustering and internalization into myeloid cells[29], which suggests that an ADC could also deliver payload to myeloid cells in an antigen-dependent manner. Thus, in principle, a tumor cell-directed STINGa ADC could deliver payload and activate STING signaling in cancer cells and in tumor-resident myeloid cells, while sparing tumor-resident B and T cells as well as normal tissues.

Here, we show that tumor cell-directed STINGa ADCs activate STING in both tumor cells and myeloid cells, leading to anti-tumor innate immune responses. We demonstrate that the STINGa ADCs are internalized into myeloid cells by FcγRI, which requires ADC-binding to target antigen and FcγR. Tumor-cell-targeted STINGa ADCs induce type III IFN production, which depends on the cancer cell STING and contributes to the STING-mediated anti-tumor innate immune activity by upregulating type I IFN and other cytokines/chemokines. These results suggest that the tissue- and cell type-specific drug delivery achieved by ADCs could deliver on the promise of STING agonism as a therapeutic approach in oncology.

## Results

### Generation of STINGa ADCs
We generated a series of STINGa ADCs comprised of a STINGa conjugated to an antibody via a chemical linker (Fig. 1a, Supplementary Fig. 1a–1d). We first synthesized STINGa ADCs with wildtype (wt) human IgG1 antibodies against two internalizing antigens on cancer cells, HER2 (ERBB2) and NaPi2b (SLC34A2)[30,31]; the wt antibodies/ADCs have the ability to engage with both their target antigens on tumor cells and FcγRs on myeloid cells (Fig. 1b, Supplementary Fig. 2). To probe the effect of STING activation only in tumor cells, we introduced mutations in the antibody Fc region that abrogate FcγR interactions while retaining tumor cell (antigen) binding as well as FcRn binding[32] (Fig. 1c). To study the antigen binding dependency of ADC activity, we generated a non-binding control ADC comprised of the STINGa conjugated to an anti-RSV antibody that does not recognize any antigen in human or mice (Fig. 1c). As expected, the Fc-wt and Fc-mutant ADCs have comparable binding to target expressing cells (Supplementary Fig. 3a), and the Fc-mutant ADC did not bind to FcγRs (Supplementary Fig. 3b).

### Tumor cell-directed STINGa ADCs activate STING in myeloid cells in an antigen- and Fc-dependent manner
To study STING activation by ADCs in myeloid cells, we cultured the human monocytes, THP1-IRF3-luciferase reporter cells, on recombinant antigen-coated plates (Fig. 1d), treated them with the ADCs, and then measured luciferase activity as a read-out for STING pathway activation. Targeted ADCs with wt Fc potently induced IRF3 reporter activity, which required ADC binding to both tumor antigen and FcγR, since minimal activity was observed with Fc-mutant and non-binding control ADCs (Fig. 1e). Notably, the Fc wt targeted ADCs were ~40-100x more potent than free STINGa payload (Fig. 1e), demonstrating the benefit of active delivery into cells by the ADC. Similar results were obtained using a THP1 and cancer cell co-culture assay (Fig. 1f). Treatment of THP1 reporter cells in co-cultures with SKBR3 (HER2 + ) or OVCAR3 (NaPi2b + ) cancer cells with the Fc-wt ADCs induced IRF3 reporter activity more potently compared to the free payload (Fig. 1g). Minimal activity was observed with control and Fc-mutant ADCs (Fig. 1g) and when THP1 cells were cultured in the absence of tumor antigen (Supplementary Fig. 3c), indicating that the ADCs internalize into myeloid cells upon binding to antigens on cancer cells. No IRF3 reporter activity was seen in *STING* knock out (KO) THP1 cells in co-cultures with cancer cells, confirming that the IRF3 reporter activity is dependent on STING activation (Supplementary Fig. 3d).

To further study the internalization of the ADCs into myeloid cells, we incubated THP1 cells with FITC fluorophore-conjugated ADCs on non-coated control plates or antigen-coated plates and used a PE-anti-FITC antibody to quantitate the cell surface-bound vs internalized ADCs. Flow cytometry analysis indicated that almost all of the targeted ADC with wt Fc was detected on the myeloid cell surface in the absence of antigen (Fig. 1h, Supplementary Fig. 4a), whereas it was internalized in approximately half of the THP1 cells in the presence of antigen. The control ADC remained cell surface-bound regardless of the antigen presence, and the Fc mutant targeted ADC-treated THP1 cells were negative for both FITC and PE, confirming the lack of binding to the cells. These results indicate that ADCs bind to FcγRs via their Fc moieties regardless of antigen-binding, but internalization into myeloid cells requires antigen binding. Consistent with this result, Supplementary Fig. 4b demonstrates the efficient internalization of the targeted antibody into myeloid cells when co-cultured with cancer cells expressing an internalizing target.

### FcγRI mediates internalization of the tumor-directed STINGa ADCs into myeloid cells
Human myeloid cells express three FcγRs: FcγRI (CD64), FcγRII (CD32), and FcγRIII (CD16)[29]. To determine which FcγRs mediate ADC internalization, we first performed flow cytometry analysis of their cell surface expression levels on THP1 cells treated with the ADCs with or without target antigen. THP1 cells express FcγRI and FcγRII, but not FcγRIII, and STINGa treatment did not impact their cell surface expression (Supplementary Fig. 5a). As shown in Fig. 1i, while FcγRII expression was not significantly impacted by any of the treatments regardless of antigen presence, the FcγRI cell surface levels were reduced with the targeted ADC-wt Fc treatment in the presence of antigen, consistent with the internalization of FcγRI. Anti-FcγRI antibody bound to human FcγRI similarly with or without prior ADC incubation (Supplementary Fig. 5b), indicating that Fc-binding to human FcγRI does not significantly block the anti-FcγRI antibody-binding. These results suggested that FcγRI mediates internalization of the antigen-bound ADCs. To test this hypothesis, we generated *FCGR1* knock out (FcγRI KO) THP1 IRF3 reporter cells. Both the HER2 and NaPi2b STINGa ADCs potently induced IRF3 reporter activity in *FCGR1* wild type (FcγRI WT) THP1 co-cultures, and the activity was significantly reduced in the FcγRI KO cell co-cultures (Fig. 1j). Residual FcγRI WT cells in the KO populations (Supplementary Fig. 5c) may explain the minimal activity observed. The free STINGa payload activity was similar in FcγRI WT or KO cell co-cultures (Supplementary Fig. 5d), confirming

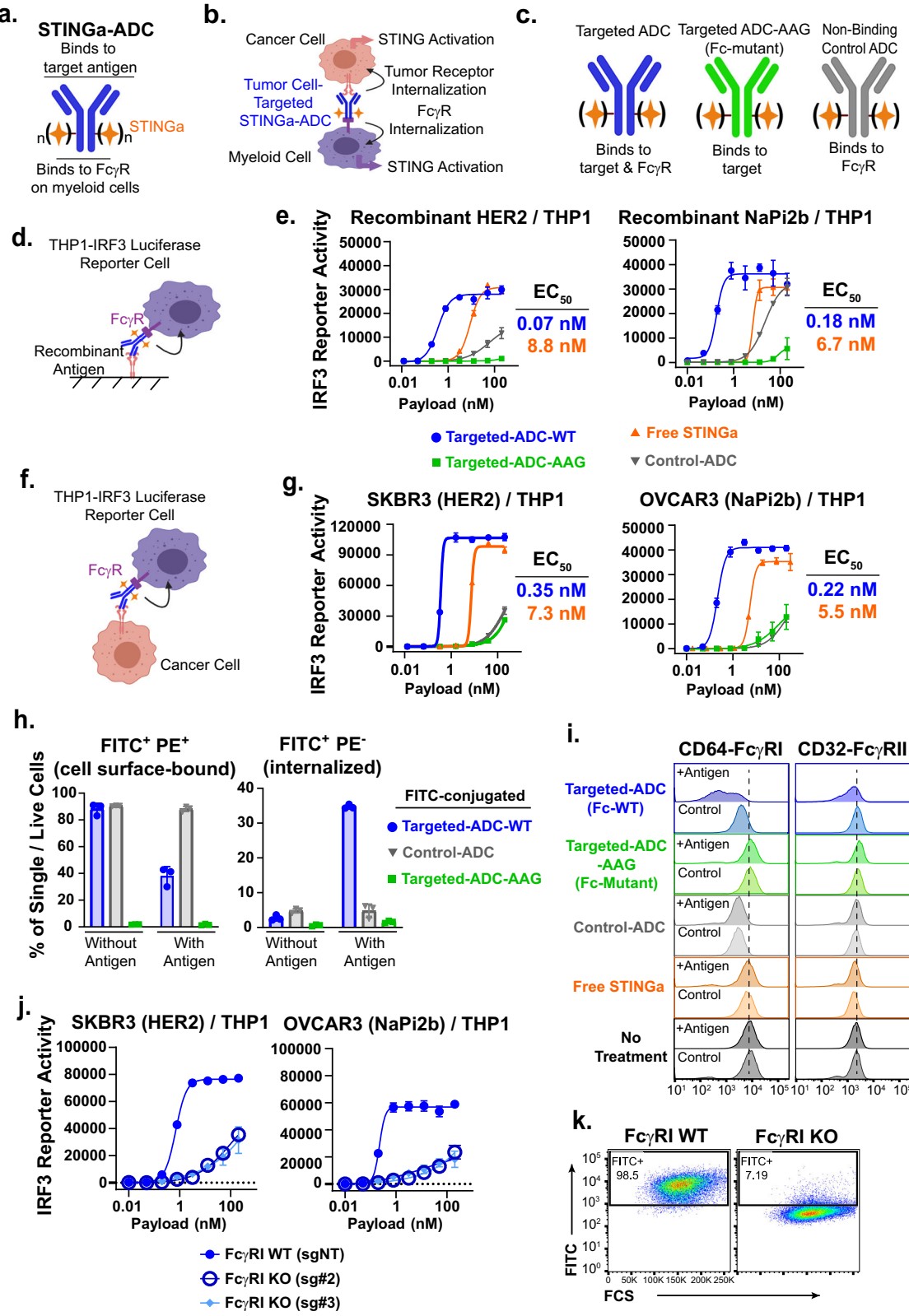

that deletion of FcγRI did not impact STING activity in these cells. Furthermore, binding of the FITC-conjugated targeted-ADC to the THP1 cells was almost completely inhibited with FcγRI deletion (Fig. 1k). Together, these data suggest that FcγRI is the primary FcγR that mediates internalization of antigen-bound ADCs into myeloid cells.

## Tumor cell-directed delivery of a STINGa via an ADC activates STING pathway in tumors with minimal systemic cytokine induction

Considering the tumor antigen-dependence of ADC internalization into myeloid cells, we surmised that the tumor cell-directed STINGa ADCs would activate the STING pathway in tumors and have minimal

**Fig. 1 | Tumor cell-directed STINGa ADCs activate STING in myeloid cells in an antigen- and Fc-dependent manner. a** Schematic of the STINGa ADC. **b** Depiction of STINGa ADC delivery into tumor cells and myeloid cells mediated by target receptor- and Fcγ receptor -binding and internalization, respectively. **c** Schematics of the ADC variants used to investigate tumor cell-intrinsic STING activation in the presence of immune cells and within the tumor microenvironment. **d** Cartoon depicting THP1-IRF3 reporter cell and recombinant antigen-coated plate assay. **e** Graphs showing IRF3 reporter activity (in relative luciferase units, mean ± SD, $n = 3$ biological replicates) in THP1 reporter cells cultured on recombinant HER2 or NaPi2b coated plates in the presence of indicated treatments (T = 24 hours). **f** Cartoon depicting THP1-IRF3 reporter cell and cancer cell co-culture assay. **g** Graphs showing IRF3 reporter activity in THP1 reporter cells co-cultured with SKBR3 (HER2) or OVCAR3 (NaPi2b) cells (T = 24 hours, mean ± SD, $n = 3$ biological replicates). **h** Flow cytometry analysis of PE-anti-FITC-stained THP1 myeloid cells, which were cultured on recombinant NaPi2b-coated plates or non-coated control plates in the presence of FITC-conjugated STINGa ADCs (20 nM based on payload) for 7 hours. Data shown are mean ± SD, $n = 3$ biological replicates. **i** Flow cytometry analysis of FcγRI or FcγRII expression (histograms) on the surface of THP1 myeloid cells cultured on recombinant NaPi2b-coated plates or non-coated control plates in the presence of FITC-conjugated ADCs (20 nM payload) for 6 hours. Data shown is representative of three biological replicates. **j** Graphs showing IRF3 reporter activity in FcγRI wild type (WT) or knock out (KO) THP1 reporter cells co-cultured with SKBR3 (HER2) or OVCAR3 (NaPi2b) cancer cells in the presence of increasing concentrations of the targeted ADCs with wt Fc (T = 24 hours). Data shown are mean ± SD, $n = 3$ biological replicates. **k** Flow cytometry analysis of FITC-conjugated NaPi2b STINGa-ADC binding to FcγRI KO THP1 cells. Data shown are mean ± SD, $n = 3$ biological replicates. All data shown are representatives of two independent experiments. Source data are provided as a Source Data File.

effect on myeloid cells in the periphery. To this end, we compared (Supplementary Fig. 6a) efficacy (tumor growth inhibition), serum cytokine induction, and tumor STING activation of a HER2-directed STINGa ADC vs. a systemically administered free STINGa (diABZI IV agonist), which is structurally similar to the ADC payload[33]. HER2-expressing SKOV3 human tumor xenografts were grown in CB.17 SCID mice that lack functional T and B cells but have intact innate immune mechanisms[34] and therefore are suitable for studying STINGa-induced innate immune responses. A single intravenous (IV) dose of 3 mg/kg (0.1 mg/kg payload) HER2 ADC led to complete tumor regressions in all mice in an antigen-dependent manner, whereas 5 mg/kg diABZI induced only modest anti-tumor activity despite being administered at a 50-fold higher dose than the ADC payload (Fig. 2a). Bodyweight loss was negligible with any treatment (Fig. 2b).

In striking contrast to the lack of anti-tumor activity, the diABZI IV agonist induced markedly high levels of systemic cytokines, while elevations were minimal in ADC-treated mice (Fig. 2c). Gene expression analysis using human vs mouse code sets revealed that the STING pathway was activated both in tumor cells (human) and host (mouse) cells (Fig. 2d), consistent with our therapeutic hypothesis. As observed in the in vitro studies, the ADC-induced changes in tumors were antigen-dependent since the control ADC did not induce significant gene upregulation. ADC-induced transcriptional changes were comparable to the free STINGa at 12 hours yet remained significantly higher at 72 hours (Supplementary Fig. 6b), which may be due to the longer half-life of ADCs compared to small molecules in vivo[33]. Indeed, pharmacokinetic (PK) analysis of the total antibody and conjugated drug concentrations in the plasma samples from the mice post HER2-ADC injection revealed parallel curves indicating long half-life and high stability of the ADC in circulation (Supplementary Fig. 6c). The STINGa ADC also exhibited significant anti-tumor activity in Balb/C mice in a syngeneic 4T1 tumor model expressing human HER2 (Supplementary Fig. 7a–7c). Taken together, these data demonstrate the antigen-dependent anti-tumor activity of STINGa ADCs, including complete and sustained tumor regressions, and indicated that the ADCs activate STING in cancer cells as well as in immune cells in vivo.

### Cancer cell-specific delivery of STINGa leads to potent anti-tumor activity

To evaluate the contribution of tumor cell-intrinsic STING activation to the anti-tumor activity of the ADCs, we utilized the Fc-mutant ADCs, which lack FcγR-binding and therefore do not activate STING in myeloid cells (Fig. 1d–g). Both HER2- and NaPi2b-targeted Fc-mutant ADCs elicited notable anti-tumor activity in the SKOV3 (HER2) and OVCAR-3 (NaPi2b) human tumor xenograft models in CB.17 SCID mice (Fig. 3a, b, Supplementary Fig. 8a–8d), which indicates a significant contribution of tumor cell intrinsic STING activation to the anti-tumor activity. The higher level of activity with Fc-wt ADCs represents the contribution of STING activation in immune cells. Neither the non-binding control

ADC nor the unconjugated antibody exhibited anti-tumor activity (Fig. 3a, b, and Supplementary Fig. 7e–7f).

To investigate the functional activity of the ADCs, we first compared cytokine induction in SKBR3 (HER2) cancer cells and primary human PBMC co-cultures using a multiplexed cytokine analysis of the culture supernatants post treatments. Both Fc-wt and Fc-mutant ADCs at 50 nM (based on payload) led to marked upregulation of several cytokines in PBMC co-cultures (Fig. 3c). Fc-wt ADC at the lower dose (1 nM) maintained significantly high levels of cytokine induction, whereas the Fc-mutant ADC was less potent. In addition, some cytokines, such as IFNγ, IL1α, IL1β, and TNFα, were detected at higher levels in the Fc-wt ADC-treated cultures, suggesting that immune cell-intrinsic STING activation is the major source of these cytokines in this case. Neither control ADC nor unconjugated antibody led to significant cytokine induction. Interestingly, the Fc-wt but not Fc-mutant HER2-ADC treatment induced cytokine production in SKBR3 and THP1 co-cultures (Supplementary Fig. 9a), indicating that the cancer cell-intrinsic STING is not activated in THP1 cell co-cultures (explored further in the next section).

We then compared ADC activity in cancer cells co-cultured with PBMCs or monocytes isolated from the same human donor. PBMCs include CD14+ monocytes, which are the main FcγRI+ population and were retained post monocyte isolation (Supplementary Fig. 9b). The Fc-wt ADCs potently induced CXCL10 production (Fig. 3d) and cancer-cell killing activity (Fig. 3e) with similar potency in both PBMC and monocyte co-cultures. The Fc-mutant ADCs elicited lower but significant activity in both co-cultures (Fig. 3d, e), consistent with their efficacy in tumors in vivo (Fig. 3a). The free STINGa payload activity was also similar in PBMC vs monocyte co-cultures, demonstrating that the FcγRI+ monocytes retain the level of innate immune responses mediated by PBMCs in the cancer cell co-cultures. The control ADC exhibited negligible activity. IncuCyte traces of cancer cell growth over time in PBMC co-cultures in the presence of the treatments are shown in Supplementary Fig. 9c. None of the treatments induced cytokine production nor impacted cancer cell viability in monocultures (Supplementary Fig. 9d, e), indicating that STING is not activated in SKBR3 and OVCAR3 cancer cell monocultures. The tumor cell-targeted ADCs also led to robust cytokine induction in the co-cultures of PBMCs and HER2-expressing HCC1954 and NaPi2b-expressing Kuramochi cancer cells (Supplementary Fig. 10a-10d), which, together with the above in vivo and in vitro data, indicates that the activity of the STINGa ADCs is observed in multiple tumor models and tumor antigens.

To further evaluate the relative contribution of the myeloid cells to the cancer cell-killing activity in PBMC co-cultures, we depleted the CD14+ monocytes from PBMCs, which retained the CD3+ lymphocytes (Supplementary Fig. 11a). HER2 ADC cancer cell-killing activity was significantly reduced in the monocyte-depleted PBMC and cancer cell co-cultures (Supplementary Fig. 11b), supporting the above findings that the myeloid cells drive robust innate immune responses to eliminate cancer cells after STINGa ADC treatment. Furthermore,

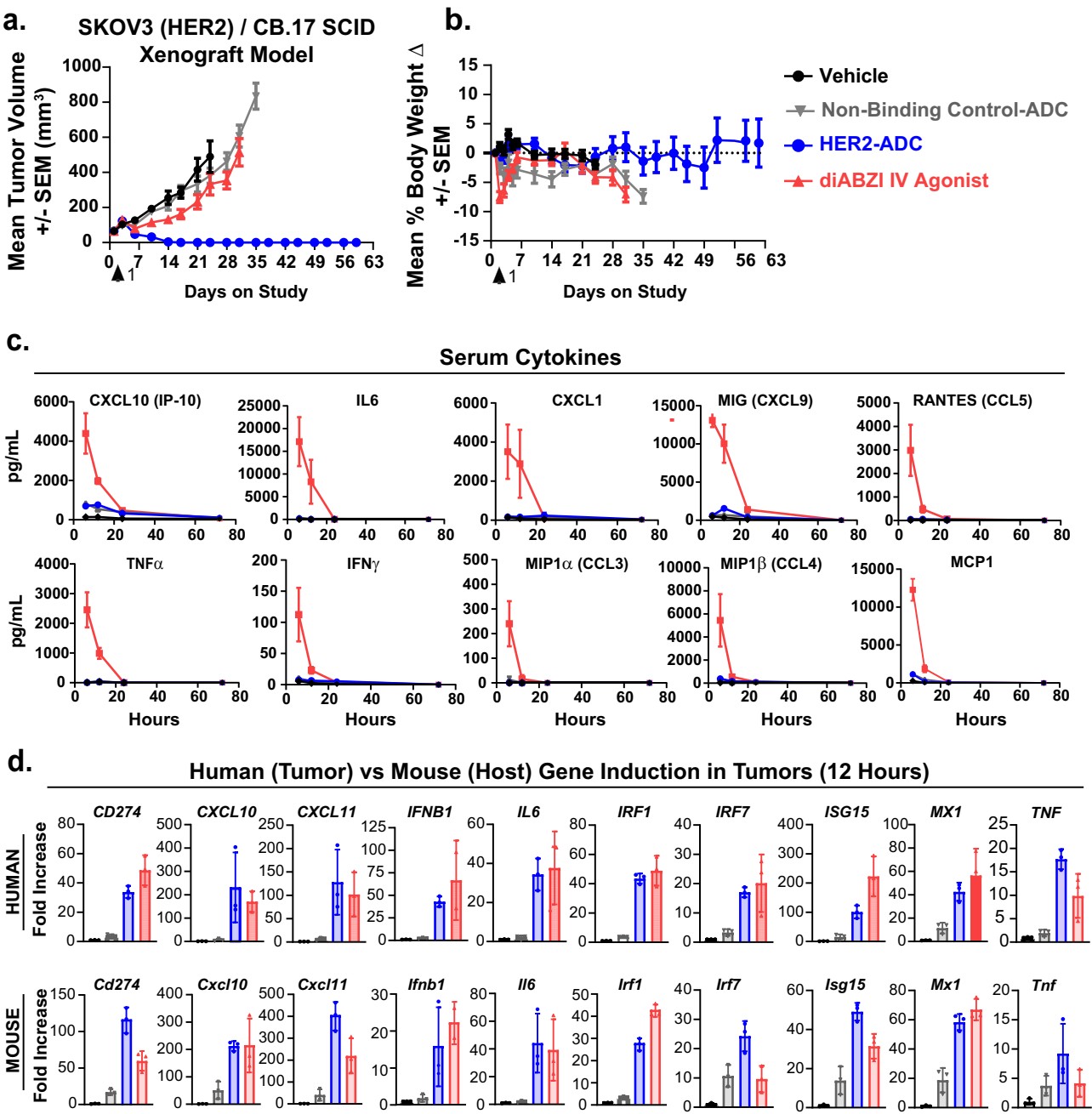

**Fig. 2 | HER2-ADC potently activates STING pathway specifically in tumors with minimal induction of serum cytokines. a** Growth of SKOV3 (HER2) human tumor xenografts in CB.17 SCID mice after a single dose of vehicle, HER2-ADC (3 / 0.1 mg/kg), non-binding control ADC (3 / 0.1 mg/kg), and the diABZI IV STING agonist (5 mg/kg). ADC doses are by antibody / payload (STINGa). Data points show mean tumor volumes ± SEM (*n* = 10). **b** Percent changes in body weights of SKOV3 tumor-bearing CB.17 SCID mice treated as described in (**a**). Each point represents the mean change in body weight ± SEM (*n* = 10). **c** Graphs showing pg/mL serum cytokine levels 6, 12, 24, and 72 hours post treatments. Data points shown are mean ± SEM (*n* = 5 mice/group). **d** Bar graphs showing fold increase in human or mouse STING pathway genes in tumors treated with the indicated test articles (T = 12 hours) as described in (**a**). Data shown are mean ± SD (*n* = 3 mice/group). Source data are provided as a Source Data File.

treatment of the cancer cell monocultures with the supernatants harvested from the cancer cell and immune cell co-cultures treated with STINGa-ADC for 24 hours led to robust killing of cancer cells (Supplementary Fig. 11c–g), suggesting that the soluble factors released in cancer cell and PBMC co-cultures downstream of STINGa-ADC treatment possess tumoricidal activity. These data suggest that the myeloid cells could drive the initial anti-tumor innate immune activity of the STINGa-ADCs.

To confirm the tumor-intrinsic STING-dependency of the Fc-mutant ADC, we assessed the cancer-cell-killing activity in *STING* wild

type (STING WT) vs *STING* knock out (STING KO) cancer cells co-cultured with PBMCs. Deletion of *STING* in SKBR3 cells (Supplementary Fig. 12a) nearly abrogated the Fc-mutant HER2 ADC cancer cell killing activity in all three single-cell clones of the STING KO SKBR3 cells in PBMC co-cultures, while minimally impacting the Fc-wt HER2 ADC activity (Fig. 3f). Treatment with unconjugated anti-HER2 antibodies with wt or mutant Fc led to a small reduction in cancer cell viability in both STING WT or KO co-cultures similarly, likely due to the cell-growth inhibitory effect of the anti-HER2 antibodies on SKBR3 cells, accounting for some of the residual activity seen with the Fc-mutant

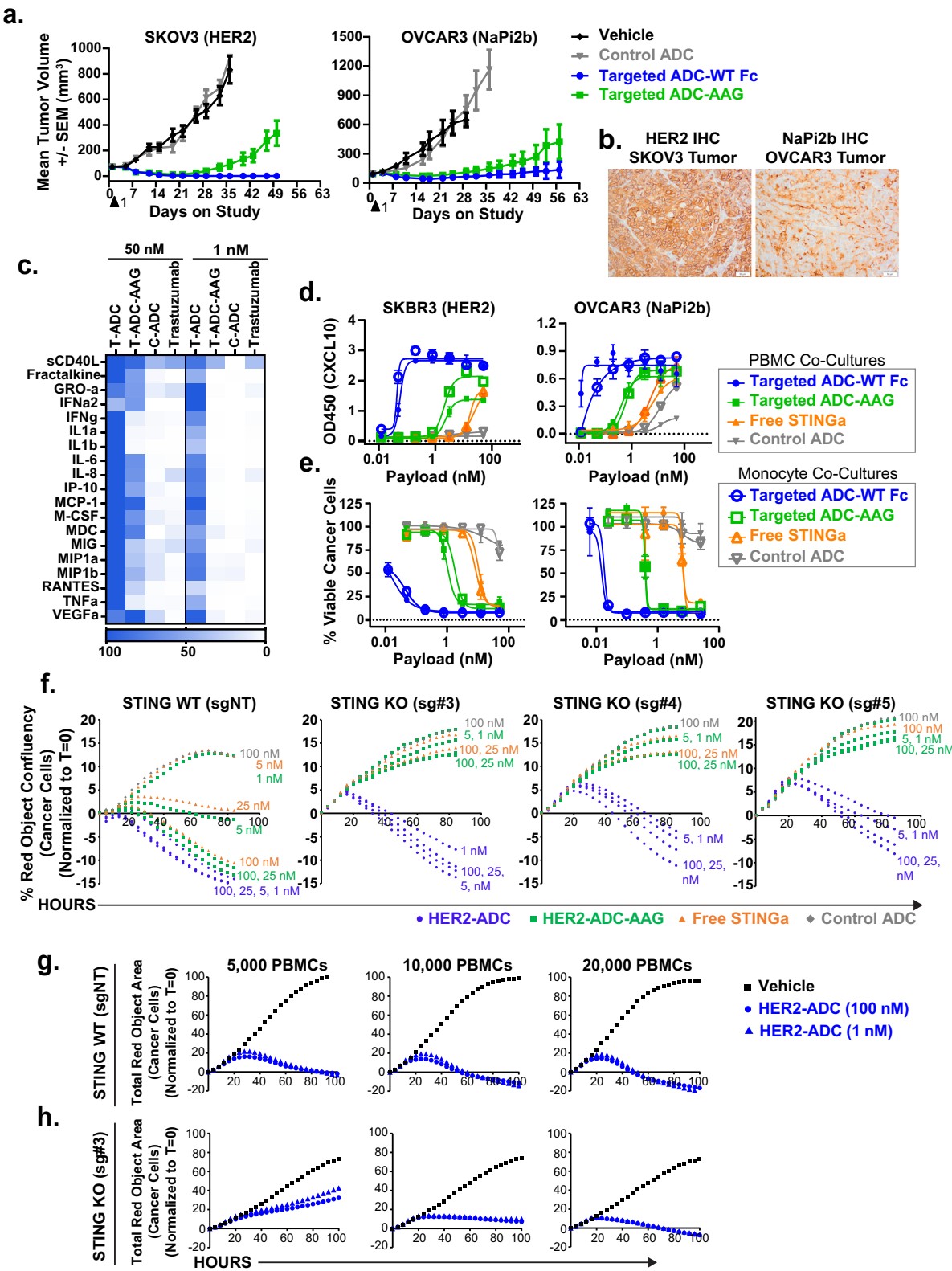

ADC in the STING KO cancer cell co-cultures (Supplementary Fig. 12b, and c). Together, these findings indicate that tumor cell-intrinsic STING pathway activation in immune cell co-cultures is capable of inducing anti-tumor innate immune activity.

We noted that the Fc-wt STINGa-ADC activity at low doses was consistently reduced in STING KO cancer cell co-cultures, suggesting that tumor cell-intrinsic STING activation could be required to maintain robust cancer-cell-killing activity at limiting conditions. To confirm this hypothesis, we co-cultured STING WT and KO cells with lower numbers of PBMCs to mimic a condition in which immune cells are sparse. Indeed, Fc-wt ADC maintained robust killing activity in STING WT cancer cell co-cultures even in the presence of a low number of immune cells (Fig. 3g). The activity in the STING KO cell co-cultures was reduced with decreasing numbers of immune cells (Fig. 3h),

**Fig. 3 | Tumor cell-specific delivery of the STINGa leads to robust anti-tumor activity in human tumor xenograft models and in cancer cell and PBMC co-cultures. a** Growth of SKOV3 (HER2) and OVCAR3 (NaPi2b) human tumor xenografts in SCID mice after a single dose of vehicle, Fc-wt HER2-ADC (3 / 0.1 mg/kg), Fc-mutant (AAG) HER2-ADC (3 / 0.1 mg/kg), and non-binding control ADC (3 / 0.1 mg/kg). ADC doses are by antibody / payload (STINGa). Data points are mean tumor volumes ± SEM (SKOV3: $n = 10$, OVCAR3: $n = 8$). **b** Representative IHC images ($n = 3$ independent tumor samples) of HER2 and NaPi2b staining in SKOV3 and OVCAR3 tumors respectively. Scale bar: 50 μm. **c** Heat map of cytokines induced in SKBR3 (HER2) and PBMC co-culture supernatants 24 hours after treatment with 50 nM (based on payload) Fc-wt HER2 ADC (T-ADC), Fc-mutant HER2 ADC (T-ADC-AAG), non-binding control ADC (C-ADC), and unconjugated anti-HER2 antibody (trastuzumab) (antibody dose equivalent of Fc-wt HER2-ADC) measured by a multiplexed cytokine assay. The scale bar shows the normalized intensity for each cytokine (average of three biological replicates) within the treatment group.

**d, e** Graphs showing CXCL10 cytokine production (T = 24 hours) (**d**) and percent viable cancer cells (T = 84 hours) (**e**) in co-cultures of SKBR3 (HER2) or OVCAR3 (NaPi2b) cells (15,000) with PBMCs (40,000) vs isolated monocytes (20,000) in the presence of indicated treatments. Data shown are mean ± SD, $n = 2$ biological replicates. **f** Graphs showing percent red object confluency as a measure of growth of STING WT or STING KO SKBR3 (HER2) NucRed cells (15,000) over time in the presence of indicated treatments in PBMC (40,000) co-cultures. **g** SKBR3 STING WT or **h** STING KO NucRed cells were treated with vehicle, 100 nM, or 1 nM HER2-ADC-wt, and the red fluorescence was traced over time in an IncuCyte instrument in the presence of 5000, 10,000, or 20,000 PBMCs. Data shown in (**f, g, h**) are the total red object confluency (cancer cells) at each time point after normalizing the average of three biological replicates to T = 0 values in each well. Results shown in (**a, b, c**) are from $n = 1$ experiment. Results shown in (**d, e, f, g, h**) are representatives of two independent experiments. Source data are provided as a Source Data File.

---

highlighting the important contributions of tumor cell-intrinsic STING to the anti-tumor activity of the ADCs, specifically in limiting conditions, such as low doses of ADCs, low antigen expression, or lower numbers of tumor resident FcγRI⁺ myeloid cells. These findings support the notion that STING activation in both myeloid cells and cancer cells contribute to the anti-tumor activity elicited by the tumor cell-targeted STINGa ADCs.

## Cues from primary human immune cells potentiate STING pathway activation in cancer cell monocultures

We sought to reconcile the lack of STING activation in cancer cell monocultures (Supplementary Fig. 9d, and e) with the significant anti-tumor activity of the Fc-mutant ADCs in vivo (Fig. 3a) and in vitro (Fig. 3c–e), which was dependent on tumor cell STING (Fig. 3f). We hypothesized that cancer cells could be enabled to activate STING in the presence of cues from primary human immune cells. Indeed, we found that most cancer cell lines failed to induce CXCL10 when treated with STINGa (Supplementary Fig. 13a), yet they became responsive to STINGa in the presence of conditioned media (CM) collected from untreated PBMC cultures (Supplementary Fig. 13b).

Based on reports of STING downregulation in tumors[20], we wondered if the defective STING signaling in cancer cell monocultures could be due to low expression of STING protein and if its expression can be induced by IFNs[35,36]. STING protein was expressed at varying basal levels across cancer cell lines and was notably induced in response to IFNβ and/or IFNγ treatments (Supplementary Fig. 13c). Moreover, SKOV3 and OVCAR3 cells responded to STINGa treatment in the presence of IFNγ as evidenced by marked increase in CXCL10 cytokine production (Supplementary Fig. 13d). We observed a modest increase in STING protein levels in these cell lines by PBMC-CM treatment (Supplementary Fig. 13e). In addition, treatment of cancer cell monocultures with STINGa in the presence of CM collected either from the primary human PBMC or isolated primary human monocyte cultures, but not THP1 malignant monocytic cell cultures, markedly induced CXCL10 cytokine production (Supplementary Fig. 13f). This result is consistent with the lack of Fc-mutant ADC activity in cancer cell and THP1 cell co-cultures and suggests that cancer cell-intrinsic STING activation requires specific factors produced by primary human immune cells in vitro. Accordingly, boiled CM did not enhance CXCL10 production in cancer cell monocultures (Supplementary Fig. 13g), demonstrating that functional factors derived from immune cells allow cancer cell-intrinsic STING pathway activation by STINGa treatment (Supplementary Fig. 13h).

## Tumor cell-directed STINGa ADCs induce type III IFNs

To further investigate the effects induced by STINGa ADCs, we performed gene expression analysis in the SKBR3 cancer cell and PBMC co-cultures after treatment with vehicle vs HER2-ADC. Consistent with the gene induction seen in SKOV3 tumors in vivo, treatment of the co-cultures with HER2-ADCs resulted in marked upregulation of a large number of genes known to be induced by STING pathway activation (Fig. 4a, and Supplementary Fig. 14a–c). We noted that *IFNL1* and *IFNL2*, members of the type III IFN family[37,38] were among the highly significantly induced mRNAs. qPCR analysis further revealed that all three type III IFNs were significantly upregulated—at comparable levels to *IFNB* mRNA – by the HER2-ADC but not Control ADC (Fig. 4b). Similarly, NaPi2b-ADC treatment of the OVCAR3 and PBMC co-cultures led to significant upregulation of the *IFNL1*, *IFNL2*, and *IFNL3* mRNA (Supplementary Fig. 15a). *IFNL1* is expressed only in humans, while *IFNL2* is expressed both in human and mice[39]. Accordingly, we confirmed the upregulation of human *IFNL1* and *IFNL2* (Fig. 4c), and mouse *Ifnl2* (Fig. 4d) mRNA in SKOV3 tumors in vivo in response to STING activation.

To demonstrate the type III IFN production at the cytokine level in co-cultures, we analyzed the culture supernatants by ELISA (detects IFNλ1, λ2, and λ3). Both Fc-wt and Fc-mutant HER2-ADCs led to marked induction of IFNλ production in SKBR3/PBMC co-cultures but not in monocultures (Fig. 4e) as seen with other cytokines downstream of STING activation. We obtained similar results with OVCAR3 and PBMC co-cultures (Supplementary Fig. 15b). To test if type III IFN is induced in cancer cells with ability to induce STING in monocultures, we used HCC1954 cells (see Supplementary Fig. 13a), which express high levels of HER2 (Supplementary Fig. 15c). Indeed, type III IFN production was induced by both Fc-wt and Fc-mutant HER2-ADC treatments in HCC1954 monocultures at levels comparable to the PBMC co-cultures (Fig. 4f), indicating that their expression pattern downstream of STING signaling is similar to that of the type I IFNs. It is important to note that the free STINGa payload also induced type III IFNs, which underscores that the ADC modality purports to achieve targeted delivery and is not expected to alter the mechanism of action inside the cell. To test if type III IFNs are regulated by TBK1/IKK signaling similar to type I IFNs in response to STING agonism, we treated the cancer cell and PBMC co-cultures with the HER2 ADC with or without TBK1/IKKε inhibitor BX795 and measured IFNβ and IFNλ cytokine production. We selected the two highest doses of BX795 that were not toxic on their own in co-cultures. The BX795 inhibited HER2-ADC-mediated induction of both IFNβ and IFNλ cytokines in a dose-dependent manner (Fig. 4g, h). Our findings reveal that type III IFNs are induced downstream of STING agonism and are regulated by the TBK1/IKKε signaling.

## Tumor cell STING is required for robust type III IFN production

Type III IFN production in immune cells has been well-established in the context of pathogen responses within epithelial surfaces[7,40,41]. To elucidate the induction of type III IFN relative to type I IFN in cancer cells in the context of PBMC co-cultures, we isolated the cancer cells after treatment of the co-cultures with HER2 ADC or Control ADC and then conducted qPCR analysis of *IFNL1* and *IFNB*. *IFNL1* and *IFNB* were

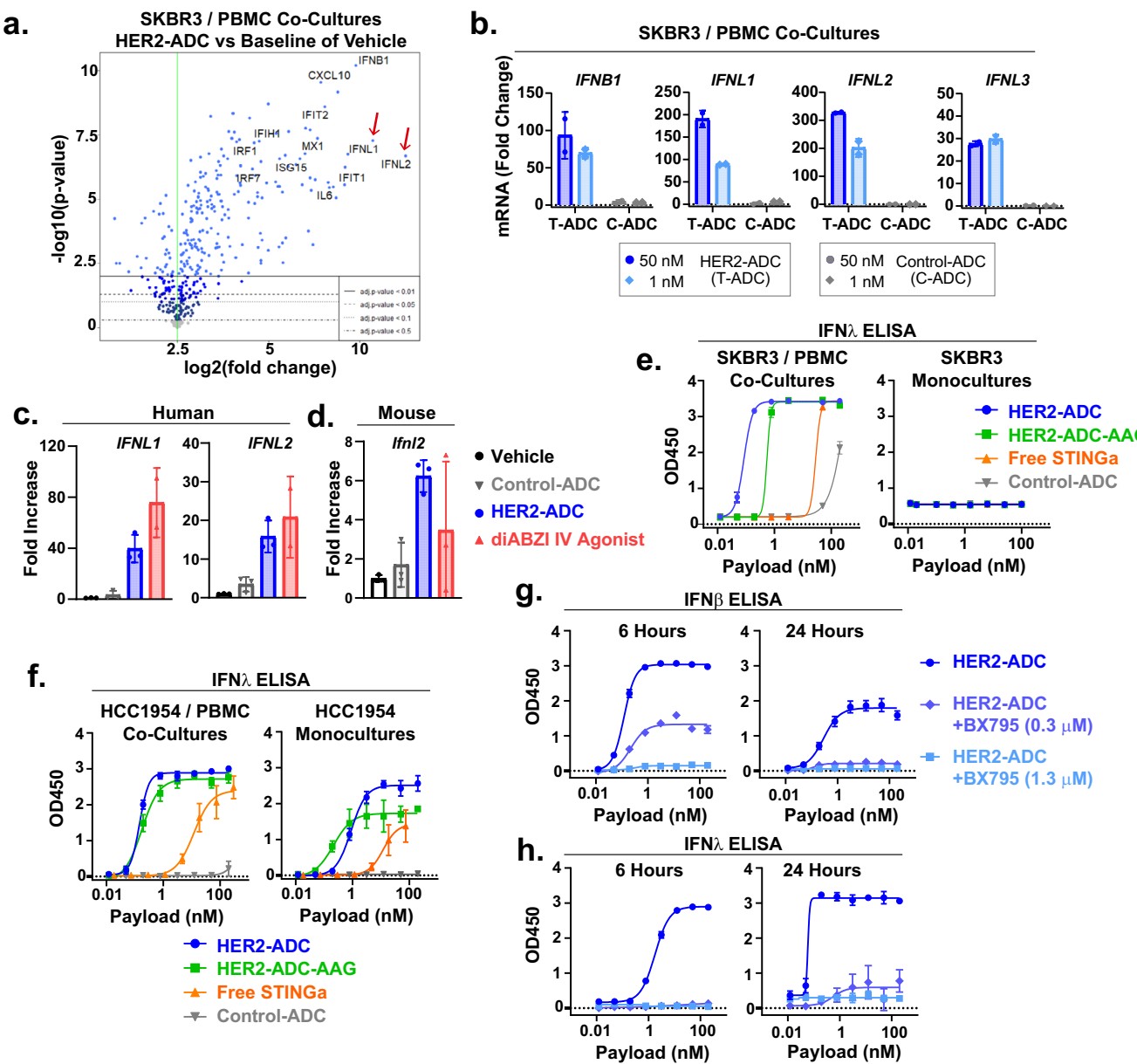

**Fig. 4 | Tumor cell-targeted STINGa ADCs induce type III IFNs. a** Volcano plots showing differentially expressed genes in SKBR3 and human PBMC co-cultures treated with vehicle or 50 nM (based on payload) HER2 ADC for 5 hours (*n* = 2 biological replicates). **b** qPCR analysis of *IFNβ*, *IFNλ1*, *IFNλ2*, and *IFNλ3* mRNA expression in SKBR3 and PBMC co-cultures treated with vehicle, 50 nM or 1 nM (based on payload) HER2 ADC (T-ADC) or Control ADC (C-ADC) for 5 hours. mRNA was normalized to GAPDH. Fold changes based on the universal RNA were calculated by the ΔΔCT method. (Mean ± SD, *n* = 2 biological replicates). **c, d** Bar graphs showing fold increase in human *IFNL1* and *IFNL2* (**c**), and mouse *Ifnl2* (**d**) in SKOV3 tumors treated with the indicated test articles (T = 12 hours) as described in Fig. 2d.

Data shown are mean ± SD, *n* = 3 mice/group. **e, f** Dose response curves for IFNλ cytokine induction in SKBR3 (**e**) or HCC1954 (**f**) and PBMC co-cultures or cancer cell monocultures treated as indicated for 24 hours. Data shown in (**e, f**) are mean ± SD, *n* = 3 biological replicates. **g, h** Dose-dependent inhibition of HER2-ADC-induced IFNβ (**g**) and IFNλ (**h**) cytokine production in SKBR3 and PBMC co-cultures by the TBK1/IKKε inhibitor BX795 (6 hours and 24 hours). Data points shown in (**g, h**) are mean ± SD, *n* = 3 biological replicates. Results shown in (**a, b, c, d**) are from *n* = 1 experiment. Results shown in (**e, f, g, h**) are representatives of two independent experiments. Source data are provided as a Source Data File.

upregulated in cancer cells at comparable levels in response to HER2-ADC but not to Control ADC (Fig. 5a, Supplementary Fig. 16a). qPCR analysis of EPCAM (epithelial cancer cell marker) and CD45 (pan-immune cell marker) confirmed the purity of the isolated cancer cells (Supplementary Fig. 16b). These data demonstrate that cancer cells robustly upregulate both type I and type III IFNs downstream of STING activation.

Interestingly, type III IFNs were induced robustly in STING WT but not in STING KO cancer cell and PBMC co-cultures by Fc wt and mutant ADCs or free STINGa (Fig. 5b). These results implied that cancer cell STING is required for a robust type III IFN production in co-cultures even with STINGa being delivered into myeloid cells. Furthermore, CD11b-ADC, which directly delivers STINGa only into myeloid cells, induced type III IFNs in STING WT but not in STING KO SKBR3 cell co-cultures (Fig. 5c), whereas CXCL10 production was not affected by STING deletion in cancer cells (Fig. 5d). Consistently, fresh human WBCs cultured on HER2-coated plates in the absence of cancer cells failed to produce type III IFNs in response to HER2-ADC treatment, despite significant levels of CXCL10 production (Fig. 5e). Together these data suggest that tumor cell-intrinsic STING is required

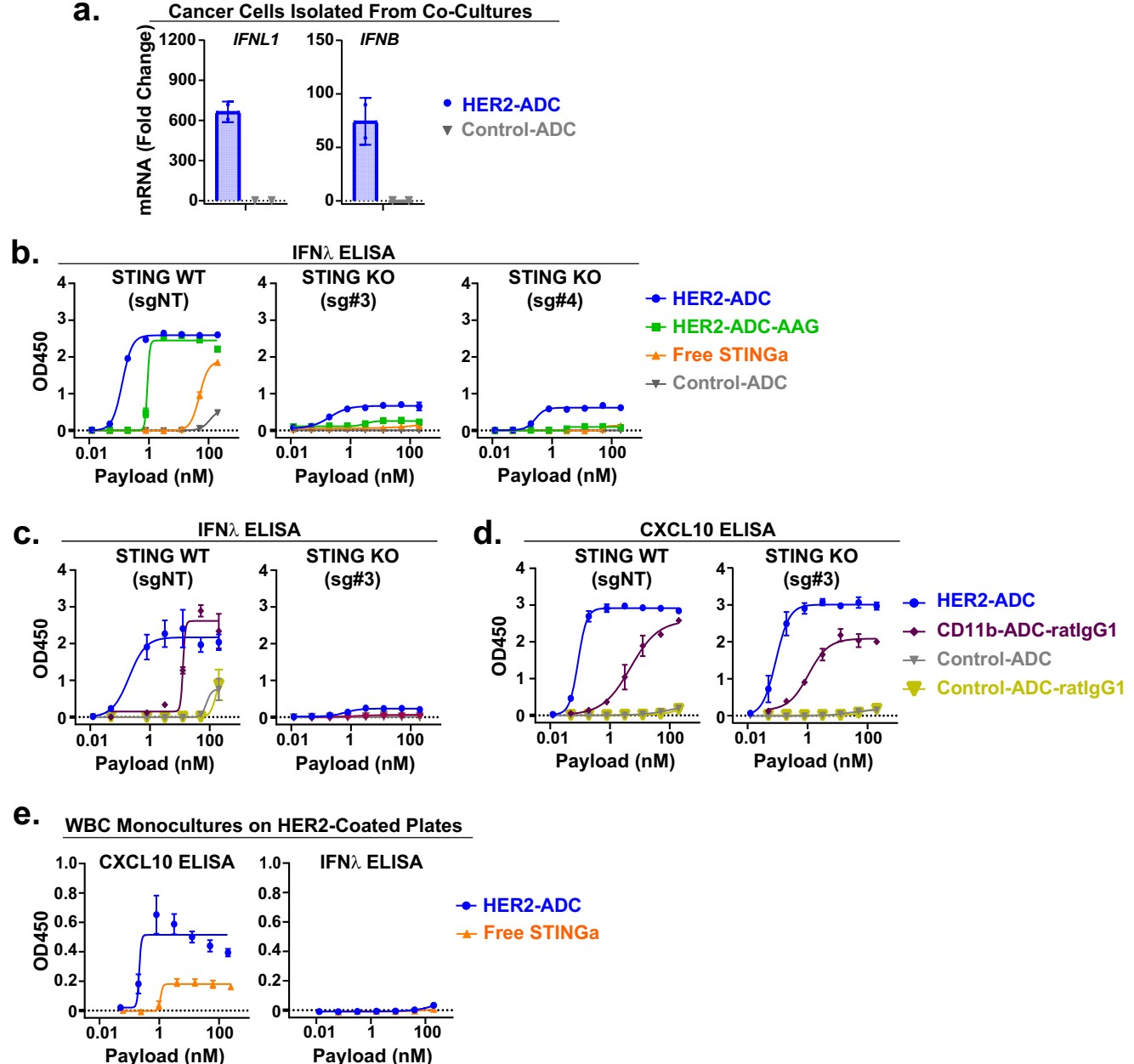

**Fig. 5 | Cancer cell-intrinsic STING is required for robust type III IFN production. a** qPCR analysis of *IFNL1* and *IFNB* mRNA expression in cancer cells isolated from the co-cultures after treatment with HER2-ADC or Control ADC (20 nM based on payload) for 5 hours. mRNA was normalized to GAPDH. Fold changes based on the universal RNA was calculated by the ΔΔCT method. Data shown are mean ± SD, $n = 2$ biological replicates. **b** Dose response curves for IFNλ cytokine induction in STING WT (sgNT) or STING KO (sg#3, sg#4) SKBR3 cells co-cultured with PBMCs for 24 hours in the presence of the indicated treatments. Data shown are mean ± SD, $n = 3$ biological replicates ($n = 2$ for C-ADC groups). **c** Dose response curves of IFNλ cytokine production in STING WT (sgNT) or STING KO (sg#3) SKBR3 cells co-cultured with PBMCs after 24 hours of treatment with HER2-ADC-wt, CD11b-ADC, and the non-binding Control ADCs with corresponding isotypes. **d** Dose response curves of CXCL10 cytokine production in the same cultures as described in (**c**). Data shown in **c**, **d** are mean ± SD, $n = 3$ biological replicates ($n = 2$ for C-ADC-ratIgG1 groups). **e** Dose response curves of CXCL10 and IFNλ cytokine production in fresh WBCs (white blood cells) cultured on recombinant HER2-antigen-coated plates in the absence of cancer cells in response to HER2-ADC and free STINGa treatments for 24 hours. Data shown are the mean ± SD, $n = 3$ biological replicates. Results shown in (**a–e**) are representative of two independent experiments. Source data are provided as a Source Data File.

for a robust type III IFN induction in co-cultures in response to STING agonism, even in the presence of intact STING signaling in immune cells.

**Type III IFN contributes to anti-tumor innate immune responses**
Type III IFNs, akin to type I IFNs, are induced in response to pathogen invasion and activate similar anti-viral and immunomodulatory gene expression programs[39]. Given the strong induction of type III IFNs in

response to STING activation in cancer cell and immune cell co-cultures and within the TME in vivo, we speculated that type III IFNs might play a role in the antitumor activity of the tumor cell-directed STINGa ADCs. Indeed, IFNλ1-neutralizing antibodies countered the cancer cell-killing activity induced by HER2-ADC, as evidenced by a ~ 6-fold increase in the EC$_{50}$ (Fig. 6a). IFNλ2-neutralizing antibodies had no effect in this assay. Of note, anti-IFNλ1 antibody is cross-reactive with IFNλ2 and IFNλ3, while the anti-IFNλ2 antibody is cross-reactive with

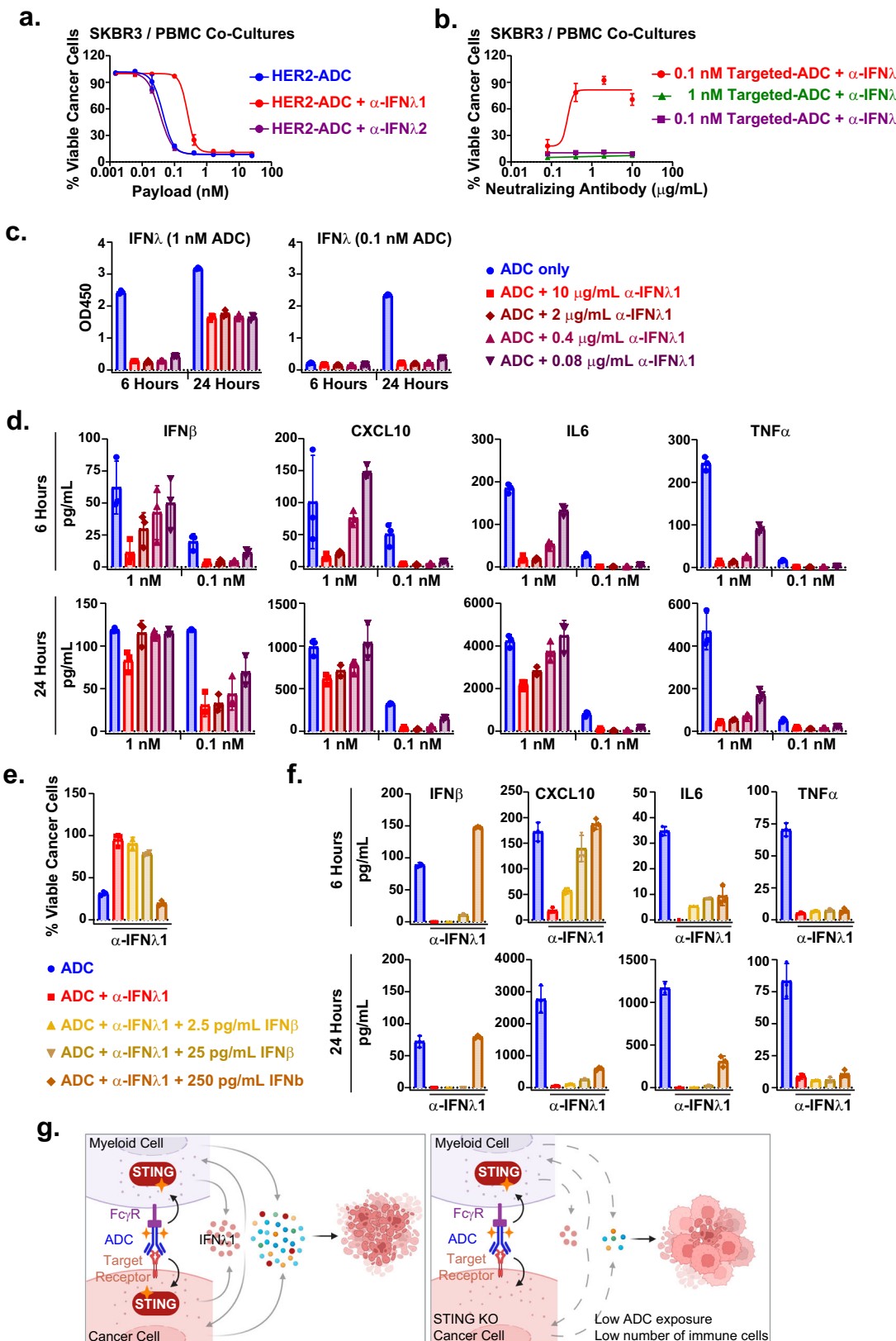

IFNλ3. Therefore, these data suggest that IFNλ1 may be the key type III IFN mediating the observed anti-tumor activity downstream of STING agonism.

Based on these data, we selected 1 nM and 0.1 nM doses (by payload) of HER2-ADC (dose range shifted by the anti-IFNλ1 antibody in the killing assay) to test if the IFNλ1-neutralizing antibodies can inhibit

the killing activity of the HER2-ADCs in a dose-dependent manner. IFNλ1-neutralizing antibodies markedly reduced the cancer cell-killing activity induced by 0.1 nM but not by 1 nM HER2-ADC treatment (Fig. 6b). IFNλ2-neutralizing antibodies did not impact the HER2-ADC killing activity in line with data shown in Fig. 6a. Consistently, the IFNλ cytokine levels were efficiently reduced by the IFNλ1-neutralizing

**Fig. 6 | Type III IFN contributes to the anti-tumor activity of STINGa ADCs in cancer cell and PBMC co-cultures. a** Dose response curves for the viability of SKBR3 NucRed cells and PBMC co-cultures treated with HER2-ADC, with or without IFNλ1- or IFNλ2-neutralizing monoclonal antibodies (2 μg/mL, T = 84 hours). **b** Viability of SKBR3 NucRed cells and PBMC co-cultures after 84 hours of treatment with 1 nM or 0.1 nM HER2 ADC with or without 10, 2, 0.4, 0.08 μg/mL IFNλ1−neutralizing antibodies. A set of wells were treated with IFNλ2−neutralizing antibodies as control. **c** IFNλ cytokine induction in the sister plates prepared as in (**b**) was analyzed by an ELISA assay at 6- and 24-hour time points. **d** IFNβ, CXCL10, IL6, and TNFα cytokines were measured in the same supernatants as in (**c**) using a multiplex cytokine assay. **e** SKBR3 NucRed cell viability in PBMC co-cultures treated

with 0.1 nM (based on payload) HER2 ADC, 10 μg/mL IFNλ1-neutralizing antibodies, and increasing doses of recombinant human IFNβ (T = 84 hours). **f** IFNβ, CXCL10, IL6, and TNFα cytokine induction in the supernatants of the sister plates as described in (**e**) after 6 hours and 24 hours of treatments using a multiplexed cytokine assay. **g** Schematic for the anti-tumor innate immune responses induced by the tumor cell-targeted STINGa ADCs. STING signaling is induced in both immune cells and cancer cells, leading to production of type III IFNs, which requires cancer cell-STING activation. Data shown in (**a**–**f**) are mean ± SD, n = 3 biological replicates, and representatives of two independent experiments. Source data are provided as a Source Data File.

antibodies in the 0.1 nM HER2-ADC-treated co-cultures (Fig. 6c). Notably, at 1 nM concentration of ADC, IFNλ cytokine levels were reduced by IFNλ1- neutralizing antibodies at the earlier time point (6 hours), but recovered over time, suggesting that at the higher ADC dose, the amount of IFNλ1 produced over time exceeds the capacity of the neutralizing antibodies at the concentrations tested. Cytokine analysis of the supernatants harvested 6 hours and 24 hours after treatment revealed that IFNβ, CXCL10, IL6, and TNFα were significantly reduced, consistent with the IFNλ levels (Fig. 6d). We obtained similar results in the OVCAR3 and PBMC co-culture system with the NaPi2b-ADC treatment (Supplementary Fig. 17a, and b). Collectively, these data suggest that type III IFN, specifically IFNλ1, plays an important role in the induction of IFNβ and other cytokines/chemokines downstream of STING signaling and the subsequent anti-tumor activity.

We next tested if exogenous IFNβ can overcome the inhibition of cancer-cell killing and cytokine induction activity of the ADC by IFNλ1 neutralizing antibodies. Addition of relevant doses of recombinant human IFNβ to the co-cultures in the presence of anti-IFNλ1 antibodies (10 μg/mL) rescued the killing activity observed with the STINGa ADC at the highest dose tested (Fig. 6e). Similarly, the cytokine induction activity was recovered partially with exogenous IFNβ addition to the co-cultures (Fig. 6f). To further evaluate the cancer-cell-killing activity of IFNβ relative to the STINGa ADC, we treated SKBR3 cancer cells in monocultures or in PBMC co-cultures with HER2-ADC or increasing concentrations of IFNβ. IFNβ treatment alone induced similar levels of cancer-cell killing in both monocultures and co-cultures in a dose-dependent manner, whereas the HER2-ADC induced cancer-cell killing only in the co-cultures, as expected (Supplementary Fig. 18a). Interestingly, the cytokine induction activity of the IFNβ alone at the doses that induced similar levels of cancer-cell-killing activity as with the HER2-ADC did not recapitulate the cytokine-induction profile of the ADC in co-cultures (Supplementary Fig. 18b). These data indicate that IFNβ contributes to but does not fully account for the anti-tumor activity elicited by the tumor-cell-directed STINGa ADCs.

**Tumor cell-directed STINGa ADCs induce STING pathway and type III IFNs in fresh human tumor fragment cultures**

Ex vivo fresh human tumor models resemble the in vivo characteristics of TME, including the tumor stroma and tumor resident immune cells, and therefore, have been increasingly used in recent years to study drug response mechanisms and to predict clinical responses[42,43]. We therefore extended this study to an ex-vivo patient-derived fresh human tumor fragment culture (PDTF) platform. We selected ovarian tumors based on the broad expression of NaPi2b in this tumor type[31,44] and obtained fresh tumors from two patients for our experiments as outlined in Fig. 7a. Retrospective immunohistochemical staining of the tumor sections demonstrated NaPi2b expression in both tumors (Fig. 7b, and Supplementary Fig. 19a−c). Flow cytometry analysis of the dissociated tumor fragments indicated that both tumors contained ~5% CD45⁺ immune cells (Fig. 7c).

Consistent with our results from in vitro and in vivo studies, treatment of the PDTF cultures with both Fc-wt and Fc-mutant

NaPi2b-ADCs for 8 hours led to robust upregulation of innate immune response genes at comparable levels in a target-dependent manner, as evidenced by the lack of gene upregulation by the non-binding Control ADC (Fig. 7d, and e). Free STINGa treatment at the equivalent payload dose of the ADCs did not induce gene upregulation in either tumor at this time point. Similarly, analysis of the supernatants by a multiplexed cytokine assay and the IFNλ ELISA revealed robust activation of IFNβ, CXCL10, IL6 (Fig. 7f, and Supplementary Fig. 19d) and IFNλ (Fig. 7g) by Fc-wt and Fc-mutant NaPi2b-ADCs. Interestingly, TNFα was not induced with any of the treatments, except for PMA-Ionomycin (PMA-I), which was included as a general immune cell agonist (Supplementary Fig. 19d). This is consistent with the co-culture results that showed a lack of TNFα production with only cancer cell-specific STING activation by Fc-mutant HER2-ADC (Fig. 3c). Induction of similar levels of cytokines by Fc-wt and Fc-mutant ADCs in tumor explant cultures with low proportion of immune cells suggest that cancer cell-intrinsic STING activation could significantly contribute to the observed ADC activity. Furthermore, cell death was detected with a visibly higher intensity after NaPi2b-ADC treatment compared to vehicle or Control-ADC treatment in Tumor#2 cultures (Fig. 7h). The STINGa ADCs in this experiment were fluorophore-labeled (Alexa Fluor-488) to confirm the target-specific binding of the ADCs to the tumor cells and carried a closely related STINGa payload with similar potency as that of the ADCs used in the remainder of our studies. Consistent with the mRNA and cytokine activation patterns, AF-488 fluorescence was detected only in the NaPi2b-ADC-treated cultures, demonstrating the target-specificity of the ADCs. Together, these data demonstrate that the in vitro and in vivo results with tumor cell-directed STINGa ADCs were also observed in the context of fresh human PDTF cultures ex vivo, despite the low frequency of immune cells.

## Discussion

Growing evidence indicates that the administration of free STINGa is associated with several limitations that could impede the overall anti-tumor effects of STING agonism in tumors[11,45–47]. Thus, tumor-targeted delivery approaches that allow systemic administration are increasingly being considered. Our results, comprising in vitro, in vivo, and ex vivo studies, suggest that tumor cell-directed STINGa-ADCs targeting an antigen expressed on the surface of cancer cells constitute a promising strategy to overcome these limitations and improve clinical outcomes. In this study, we have elucidated several mechanistic aspects of the anti-tumor innate immune activity of the STINGa ADCs. Our findings indicate potential benefits of this therapeutic approach, including targeted delivery to the tumor relative to the periphery, and the productive delivery of STINGa to cancer cells and FcγRI-expressing myeloid cells in an antigen-dependent manner.

We show that tumor cell-directed STINGa ADCs activate STING in cancer cells and myeloid cells following target antigen-mediated and FcγRI-mediated internalization respectively. FcγRI-mediated internalization requires ADC binding to its target antigen, and internalization into myeloid cells remains efficient even when the target antigen is known to internalize into cancer cells. The kinetics/dynamics of ADC internalization into cancer cells vs myeloid cells is likely impacted by

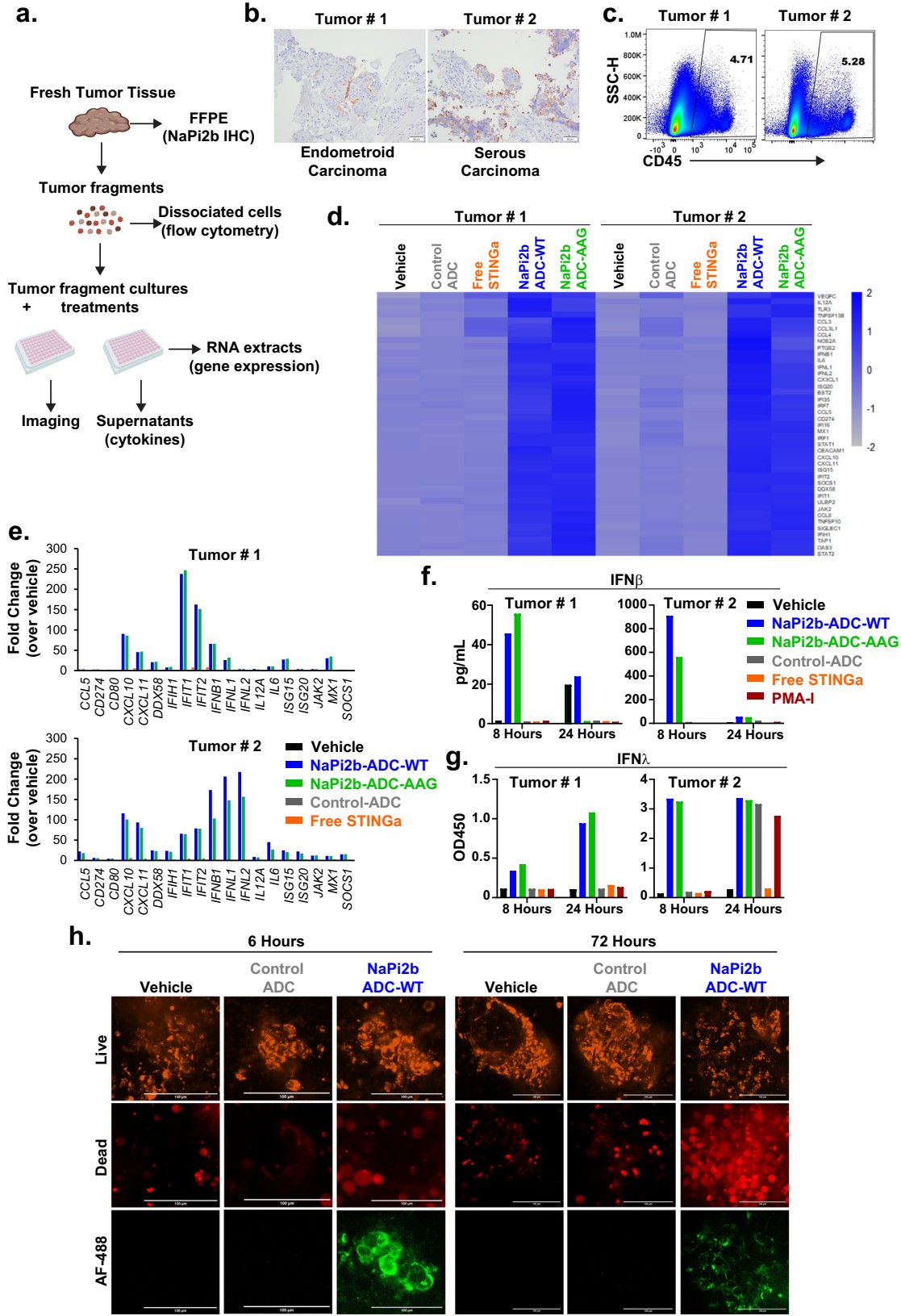

the target expression levels, ADC concentration, and receptor binding / cell surface recycling kinetics. Further studies are needed to elucidate the details of this process in tumors.

We demonstrate that tumor cell-directed delivery of a STINGa via an ADC induces complete tumor regressions in a xenograft model after systemic administration, outperforming the systemically administered diABZI free STINGa at a 50-fold lower dose. Moreover, the tumor cell-directed STINGa ADC induces much lower serum cytokine elevation compared to that of the diABZI. It will be informative to compare STINGa ADCs to free STINGa in immunocompetent mouse models including genetically engineered mouse models and ultimately in clinical trials. The ADCs bind to the FcγRs on myeloid cells

**Fig. 7 | Cancer cell-targeted STING ADCs induce STING pathway and type III IFNs in fresh human tumor fragment cultures ex vivo. a** Schematic of the experimental outline. **b** IHC images of NaPi2b staining in the ovarian tumors from two different patients included in the ex vivo assays. Scale bars: 50 µm. **c** Flow cytometry analysis of CD45 expression (immune cells) in dissociated tumors prior to ex vivo assay setup (gated on single live cells). **d** Heat map showing the normalized mRNA counts of the STING pathway genes and **e** Fold changes in select STING pathway genes in the RNA extracts of the tumor fragment cultures after 8 hours of indicated treatments over vehicle treatment. **f** IFNβ cytokine induction in the tumor fragment cultures after 8 hours and 24 hours of indicated treatments

(20 nM based on payload) measured using a multiplexed cytokine assay (CXCL10, IL6, and TNFα analysis was included in the assay and the data is shown in Supplementary Fig. 19d). **g** IFNλ cytokine induction in the same supernatants as in (**f**) were analyzed by an ELISA assay. **h** High content confocal microscopy images of PDTF cultures at 6 hours and 72 hours after the indicated treatments. Fragments from Tumor#2 were seeded in parallel with the cultures as described above and first incubated with cell viability dyes followed by treatment with vehicle, AF-488-conjugated non-binding Control ADC or NaPi2b-ADC (Fc-wt) (20 nM based on payload). Scale bars: 100 µm. Source data are provided as a Source Data File.

via their Fc moiety, raising the question of whether they can be internalized by the myeloid cells in circulation, leading to systemic inflammation. However, our studies illustrate the requirement of antigen-binding for ADC internalization and STING activation in myeloid cells, thus explaining the low levels of STING activation in the periphery in vivo.

Exploiting the STING pathway to enhance anti-tumor T cell responses has been well-documented in a plethora of preclinical studies[8,11]. Recently Wu et al. reported that an EGFR-directed STINGa ADC induces adaptive immune responses in syngeneic mouse models and synergizes with CPI combination therapy[48]. In line with the literature, we show that the HER2-directed STINGa ADC exhibits anti-tumor activity in a HER2-expressing syngeneic mouse model. Furthermore, our studies uncover a potent anti-tumor activity directly mediated by STING-induced innate immune responses, which has been overlooked in the field. We demonstrate that the tumor cell-targeted STINGa ADCs induce type I IFN and innate immune pathways and elicit anti-tumor activity in human tumor xenografts in SCID mice, which lack functional T and B cells, but retain an intact myeloid compartment. These results suggest that innate immune responses immediately downstream of STING activation are the primary driver of the anti-tumor activity of STINGa ADCs. Likewise, the results of our in vitro co-cultures with human cells support this concept: STINGa ADCs potently induce similar levels of cytokines and killing of cancer cells in co-cultures with PBMCs vs isolated monocytes, which are the FcγRI-expressing myeloid cell population within PBMCs. Furthermore, the supernatants collected from co-cultures following STINGa-ADC treatment, induce direct killing of cancer cells in monocultures, indicating a tumoricidal activity. Thus, potent activation of STING signaling and innate immune responses by the tumor cell-targeted STINGa ADCs have the potential to drive efficacy in cancers, including those resistant to T cell-mediated elimination, in addition to stimulating anti-tumor NK cell or T cell activity and inducing long-term immune memory, particularly in combination with CPI treatments.

A key mechanistic finding revealed by our studies is the requirement of tumor cell-intrinsic STING activation for a robust type III IFN production in response to STING agonism. Type III IFNs are critical players in pathogen responses specifically in epithelial tissues; however, their role in anti-tumor immunity has not been well-studied[39,49,50]. The signaling cascades and gene expression profiles induced by type III IFNs, in part, overlap with those of type I IFNs. Both types I and III IFNs are broadly expressed across cell types, yet the signal strength/duration by which they are induced, the kinetics of their expression, and the expression pattern of their receptors on different cell types/tissue distinguish their biological functions[49]. In our co-culture system, *IFNB* and *IFNL1* mRNA expression were induced by tumor cell-targeted STINGa ADCs downstream of TBK1 signaling. Strikingly, type III IFN but not type I IFN production required STING activation in cancer cells. It is unclear how cancer cell STING regulates overall type III IFN production in response to STING agonism. It is plausible that crosstalk between cancer cells and immune cells, which may involve other key factors as a result of cancer cell STING activation, is required for robust levels of type III IFN production. In addition to the co-culture setting, we observed strong type III IFN induction along with type I IFNs following

STINGa-ADC treatment of fresh human tumor cultures ex vivo. We propose that tumor-cell STING activity and type III IFN-production capacity could potentially serve as biomarkers for patient stratification strategies. Therefore, future studies addressing the interplay between cancer-cell STING signaling and type III IFN production in the TME, as well as how IFNλ1 mediates STING-induced anti-tumor activity, could be highly insightful. Of note, the utility of mouse studies for this purpose will be limited since mice do not express IFNλ1.

Our data indicate that type III IFNs, specifically IFNλ1, are required for type I IFN production and the anti-tumor activity elicited by cancer-cell-targeted STINGa ADCs. IFNλ1-neutralizing antibodies inhibited the STINGa-ADC-mediated cancer cell-killing activity and induction of IFNβ and other cytokines/chemokines in cancer/immune cell co-cultures, which was partially rescued by the addition of recombinant IFNβ protein. These findings support the hypothesis that IFNβ in part mediates the anti-tumor innate immune activities in response to STINGa-ADC treatment, downstream of IFNλ1. Interestingly, the cancer cell-directed STINGa ADC (which delivers STINGa into both immune cells and cancer cells) retains its potent cancer-cell-killing activity in STING KO cancer cells in co-cultures with immune cells at high doses, but its activity is reduced at low doses or when immune cells are sparse. This suggests that under high exposure conditions, strong activation of STING in immune cells bypasses the requirement of cancer cell STING and IFNλ1 for IFNβ expression and innate immune activation. We speculate that under limiting conditions, such as low doses of the STINGa ADC, low target expression on cancer cells, or lower frequency of FcγRI-expressing myeloid cells in the TME, which are expected to result in weaker induction of type I IFNs, type III IFN and possibly other factors/signals contributed by the cancer cell-intrinsic STING activation may provide a boost to increase the levels of type I IFN and ISGs, resulting in stronger anti-tumor innate immune activity. Indeed, the preferential expression of type III IFNs following epithelial infections by RNA or DNA viruses has been previously demonstrated[40,51-53]. Thus, type III IFN has been proposed as the main mechanism of protection against viral infections in epithelial tissues while the type I IFN responses would ensue in the case of high levels of viral RNA-sensing due to uncontrolled viral replication[49].

In summary, we have demonstrated the therapeutic rationale for tumor cell-directed delivery of a STINGa via an ADC and have shown that STING activation in cancer cells and myeloid cells both contribute to the anti-tumor activity. We discovered that the STINGa treatment induces type III IFNs in the TME and that tumor cell STING is required for robust type III IFN production, which in turn regulates type I IFN and other cytokines/chemokines for efficient anti-tumor responses to STING agonism as summarized in our proposed model (Fig. 6g). These findings highlight a critical role for tumor cell-intrinsic STING in the anti-tumor innate immune responses mediated by STING agonism and provide the rationale for tumor cell-targeted STINGa ADCs as an effective immunotherapy strategy.

## Methods
### Ethics statement
All studies in this paper comply with all relevant ethical regulations. In vivo mouse studies were approved by the Institutional Animal Care

and Use Committee (IACUC) of Charles River Discovery Services (CRL; North Carolina, USA) and Translational Drug Development, LLC (TD2; Arizona, USA). CRL and TD2 are accredited under the Association for Assessment and Accreditation of Laboratory Animal Care International (AAALAC). All fresh human tumor tissue samples were appropriately consented, and the studies were approved by the IRB (Ohio State Biomedical Institutional Review Board).

## Cell lines and culture conditions
HCC1954 (ATCC; CRL-2338), MDA-MB-175-VII (ATCC; HTB-25), Kur-amochi (JCRB; #JCRB0098), and 4T1 (ATCC; CRL-2539) cells were cultured in RPMI 1640 with 10% FBS and 1% Penicillin/Streptomycin. SKBR3 (ATCC; HTB-30), SKOV3 (ATCC; HTB-77), MDA-MB-475 (ATCC, HTB-131), and JIMT-1 (DSMZ; ACC589) cells were cultured in DMEM with 10% FBS and 1% Penicillin/Streptomycin. Calu-3 (ATCC; HTB-55) cells were cultured in EMEM with 10% FBS and 1% penicillin/strepto-mycin. OVCAR3 (ATCC; HTB-161) cells were cultured in RPMI 1640 with 20% FBS, 1% Penicillin/Streptomycin. THP1-Dual (Cat# thpd-nfis) and THP1-Dual KO-STING (Cat# thpd-kostg) reporter cells were purchased from Invivogen and cultured according to vendor instructions. Cells were routinely tested for mycoplasma contamination and authenti-cated using short tandem repeat analysis on a quarterly basis (IDEXX BioAnalytics). All cells used in this study were negative for myco-plasma. Primary human PBMCs (frozen) were purchased from STEM-CELL Technologies (Cat# 70025.2). White blood cells (WBCs) were isolated from fresh human blood (STEMCELL Technologies, Cat# 70508.2) by red blood cell lysis using ammonium chloride solution (STEMCELL Technologies, Cat# 07800).

## Generation of NucRed cancer cells
SKBR3 and OVCAR3 cells stably expressing nuclear restricted mKate fluorescent red protein were generated by transduction with Incu-Cyte© NucLight Red Lentivirus reagent (Sartorius, Cat# 4476). Stably transduced cells (designated as SKBR3- or OVCAR3-NucRed cells) were selected in puromycin-containing media (2 μg/mL) for 2–3 days and expanded in their respective culture medium.

## Generation of 4T1-human HER2-expressing mouse cancer cells
4T1 mouse cancer cells expressing human HER2 were generated by transduction with Lentivirus reagent (GeneCopoeia, Cat# LPP-Z2866-Lv105). Stably transduced cells were selected in puromycin-containing media (5 μg/mL) for 2-3 days and expanded in culture medium (DMEM, 10% FBS). 4T1-human HER2-transduced cells expressing HER2 on the cell surface were enriched by FACS using anti-HER2 antibody staining method as described in the Cell Binding Assays section below. 4T1-human HER2 cells were derived from the commercially available cell line and materials, which are subject to certain restrictions. Please see the provider website for details (https://www.atcc.org/products/htb-30#product-permits). The authors can offer assistance with questions on the protocols for the derivation of these cells from commercially available materials.

## Antibodies
The following antibodies were used in the ADC generation or as unconjugated antibody controls: Anti-HER2 antibody Trastuzumab biosimilar was purchased from STC Biologics (Cat# STC101), CD11b antibody was purchased from BioXcell (Cat# BE0007) to generate CD11b-STINGa-ADC. Recombinant antibodies including anti-NaPi2b human IgG1 (XMT-1535)[54], anti-RSV human IgG1 (Palivizumab) used to generate non-binding Control ADC, and Fc-engineered antibodies were produced by a contract research organization using standard techniques. Briefly, expression vectors containing the heavy chain and light chain coding sequences of the above-mentioned antibodies were transfected in HEK or CHO cells and purified by Protein A affinity chromatography. The resulting recombinant human IgG1 antibodies

were tested to ensure a purity of >95% using SDS-PAGE and analytical size exclusion chromatography (SEC). Fc mutant antibodies with Fc regions engineered to abrogate FcγR binding were designed with three mutations in the heavy chain constant region L234A, L235A, and P329G (AAG; Kabat Eu numbering)[32] and generated through standard mole-cular biology procedures by a contract research organization. All antibodies used for ADC generation are well-established and validated to bind to their targets with expected affinities using sandwich ELISA-based assays at Mersana Therapeutics. Their purity and quality (lack of aggregation) were confirmed by LC-MS and SEC. Validation of the Biolegend antibodies is shown on the corresponding product pages on the BioLegend website (traceable from catalog numbers). NaPi2b antibodies are licensed from a third party and are subject to certain restrictions. The authors can offer assistance with questions on the protocols for the synthesis of these antibodies and ADCs.

## STING agonist and ADC synthesis
The STING agonist (STINGa) and the STINGa scaffold-linker (reagent required for conjugation of the payload to the antibody) were pre-pared as previously described (compound 11 in US Patent No. 11,155,567, issued Oct 26, 2021, and compound 7 in PCT Patent Appli-cation WO2021202984A1, published Oct 7, 2021, respectively). All ADCs were prepared in a similar fashion [PCT Patent Application WO2021202984A1, published Oct 7, 2021]. Briefly, ADCs were gener-ated by conjugating maleimide-containing scaffold-payload to native cysteines exposed by the reduction of interchain disulfides. Antibodies were reduced at 5 mg/mL in 50 mM HEPES, 1 mM EDTA, and pH7 using 4−8 equivalents of TCEP-HCl and reacted for 90 min at 37 °C. Maleimide-containing STINGa scaffold-linker was solubilized in N,N-dimethylacetarnide (Sigma, Cat# 185884) and then 8 to 12 molar equivalents were added to the reduced antibodies for a final reaction concentration of 9% N,N-dimethylacetamide and incubated for 60 min at 37 °C. The conjugations were quenched with 15 molar equivalents of L-cysteine for 45 min at room temperature. Crude ADCs were purified by a step gradient using ceramic hydroxyapatite resin (CHT type II 40 μm, Bio-Rad, Cat# 1584200) following the manufacturer's recom-mendations. Final ADCs were tested by hydrophobic interaction chromatography (HIC) (Supplementary Fig. 1C), size exclusion chro-matography (SEC) (Supplementary Fig. 1D), and UV-vis spectro-photometry. DAR and concentration were determined by UV-Vis spectrophotometry using a NanoDrop2000 (Thermo-Scientific) and measuring absorbance at 280 nm and 320 nm[55]. HIC was performed on a TSKgel Butyl-NPR column (2.5 μm, 4.6 mm × 100 mm, Tosoh Bioscience, Cat# 0042168) at 35 °C and eluted with a 25 min gradient from 0 to 100% B at a flow rate of 1 mL/min (mobile phase A: 1.5 mol/L ammonium sulfate in 25 mmol/L sodium phosphate, pH 7; mobile phase B: 25 mmol/L sodium phosphate, pH 7, 10% isopropanol). SEC was tested on a TSKgel G3000SWXL (5 μm, 7.8 mm × 300 mm, Tosoh Bioscience) at 35 °C using isocratic conditions at a flow rate of 0.75 mL/min for 25 min (mobile phase 25 mmol/L sodium phosphate, 150 mmol/L sodium chloride).

## Fluorophore labeling of ADCs
Fluorescein (FITC) labeled antibodies or ADCs were prepared using NHS-ester Fluorescein (Thermo Fisher, Cat# 46409). Each ADC or antibody was reacted with 15 equivalents of FITC (2.5 mg/mL in dimethyl acetamide) at an antibody concentration of 5 mg/mL in 50 mM HEPES, 1 mM EDTA, pH7. The reactions were rocked overnight at room temperature. Excess FITC was washed with 6 rounds of ultrafiltration followed by 10-fold dilution using a 30 kDa MWCO centrifugal filter (Millipore-Sigma, Cat# UFC503008). FITC-labeled-ADCs/antibodies were characterized by UV-Vis spectroscopy, size exclusion chromatography, and hydrophobic interaction chromato-graphy. Alexa-Fluor-488 (AF-488) labeled ADCs were prepared using an AF-488 NHS ester (ThermoFisher, Cat# A20000). The non-binding

Control ADC or NaPi2b ADC (Fc-wt) was reacted with 11 equivalents of AF-488 (1 mg/mL in dimethyl acetamide) at an ADC concentration of 4 mg/mL in 50 mM HEPES, 1 mM EDTA, pH 7. The reactions were rocked for 1 hour at room temperature. Excess AF-488 was washed with 6 rounds of ultrafiltration followed by 10-fold dilution using a 30 kDa MWCO centrifugal filter (Millipore-Sigma, Cat# UFC503008). AF-488-labeled-ADCs were characterized by UV-Vis spectrophotometry, SEC, and HIC.

## THP1 IRF3 reporter cell assays

For the plate-bound recombinant antigen assay, plates were first coated overnight with 1 μg/mL recombinant human HER2 protein (Sino Biological, Cat# 10004-H08H4). The following day, plates were washed with 0.1% TBST and blocked with 3% BSA in PBS (all solutions filter-sterilized). After washing with assay medium 3x fresh assay medium containing treatments (200−0.01 nM based on payload, 1:4 dilution) and 50,000 THP1 reporter cells/well were added to the plates and incubated for 24 hours at 37 °C, 5% $CO_2$. For cancer cells and THP1 IRF3 reporter cell co-culture assays, SKOV3 cancer cells were seeded in 96 well tissue culture plates ( ~ 15,000 cells/well) and allowed to attach overnight. The culture medium was replaced with assay medium (RPMI-1640, 10% FBS, 1% penicillin/streptomycin), and after adding the indicated test articles, the plates were incubated for 20 min at 37 °C. THP1 Dual IRF3 reporter cells (50,000 cells/well) were added and the plates were incubated for 24 h at 37 °C, 5% $CO_2$. For both co-cultures and plate-bound antigen assays, supernatants were assayed for luciferase activity using ANTI-Luc luminescence assay reagent (Invivogen, Cat# rep-qlc) on a SpectraMax M5 plate reader. Dose-response curves for all assays were generated using GraphPad Prism software (version 9.3.1). $EC_{50}$ values were determined from a four-parameter curve fitting in GraphPad Prism.

## Cancer cell / PBMC co-cultures and IncuCyte cancer cell killing assay

Cancer cell death in monocultures or in co-cultures with immune cells was determined using an IncuCyte cancer cell killing assay. Cancer cells stably expressing mKate fluorescent red protein were seeded in 96 well tissue culture plates and allowed to attach overnight. The following day culture medium was replaced with fresh assay medium (RPMI-1640, 10% FBS, 1% penicillin/streptomycin) containing treatments (200−0.01 nM based on payload, 1:4 dilution) and incubated for 20 minutes at 37 °C, followed by addition of PBMCs (1:2−1:3 ratio). Plates were then placed in an IncuCyte live cell imaging instrument in an incubator (37 °C, 5% $CO_2$) and scanned every 4 hours over 3−4 days. Red object confluency or area (cancer cells) over time was quantified using IncuCyte Zoom (version 2015 Rev1) or IncuCyte S3 (version 2020 A) software. Percent viable cells were calculated relative to the average of the red object confluency/area of control wells. Dose-response curves were generated using GraphPad Prism software (version 9.3.1). $IC_{50}$ values were determined from a four-parameter curve fitting by the Prism.

## Generation of *STING* knock out SKBR3 Cells and *CD64* knock out THP1 cells

CRISPR/CAS9 was used to generate gene knock-out cell lines. For *STING* knock out, SKBR3 cells were transfected with a non-targeting sgRNA and three different sgRNAs targeting the human *STING* or using the TrueGuide™ Synthetic gRNA, TrueCut™ Cas9 Protein v2, and Lipofectamine™ CRISPRMAX™ Transfection Reagent from Thermo Fisher according to the manufacturer's protocol. sgRNA sequences were: sgNT (non-targeting): AAAUGUGAGAUCAGAGUAAU; for *STING* knock out: sg#3: TACTCCCTCCCAAATGCGGT; sg#4: CTCGCAGGCA CTGAACATCC; and sg#5: GTTAAACGGGGTCTGCAGCC. Seven days post-transfection, single cells were sorted in 96-well plates containing DMEM with 20% FBS and 1% penicillin/streptomycin, and clones were formed in 2-3 weeks (media was refreshed once-twice a week). Several single-cell clones were expanded, and *STING* deletion was confirmed

by Western blot analysis. *STING* knock-out cells were designated as STING KO. The single cell clones with no STING protein expression were transduced to stably express the nuclear restricted mKate fluorescent protein as described above. *CD64* knock-out (CD64 KO) THP1 Dual cells were generated similarly. sgRNA sequences used: sgNT (non-targeting): AAAUGUGAGAUCAGAGUAAU; for *CD64* knock out: sg#2: GCAAGGUUACGGUUUCCUCU; sg#3: UCUACACAGUGGUUUCU CAA. 5-7 days post-transfection, CD64 KO THP1 cells were isolated by FACS based on negative CD64 cell surface expression and expanded in culture. SKBR3 STING KO and THP1 STING KO were derived from commercially available cell lines and materials, which are subject to certain restrictions. Please see the provider website for details (https://www.invivogen.com/terms-conditions#anchor-cell) (https://www.atcc.org/products/htb-30#product-permits). The authors can offer assistance with questions on the protocols for the derivation of these cell lines from commercially available materials.

## Western blot analysis

Cells were treated in 6-well plates. At the end of the treatments, culture medium was aspirated, wells were carefully washed with ice-cold PBS and lysed in m-PER mammalian cell lysis reagent (Thermo-Fisher, Cat#78501). Lysates were scraped, transferred to a chilled Eppendorf tube and passed through a 23Ga needle using a syringe, followed by centrifugation (300xg, 15 minutes, 4 °C). Protein concentrations in the supernatants were determined using a BCA assay (Pierce, Cat# 23225). After adding sample loading buffer (Invitrogen, Cat#AM8547), lysates were denatured at 95 °C for 10 min. and run on a NuPage 4−12% bis-tris gradient gel (Invitrogen, Cat# NP0322BOX) in MOPS buffer (Invitrogen, Cat# NP0001). Protein transfer was performed using nitrocellulose membranes and iBLOT2 Gel Transfer System (Thermo Fisher, Cat# IB21001). Membranes were probed using the primary antibodies: rabbit monoclonal anti-STING (Cell Signaling Technologies, Cat# 13647, 1:1000 dilution) and anti-β-Actin (Licor, Cat# 926-42212, 1:5000 dilution) and secondary antibodies: IRDye® 800CW donkey anti-rabbit IgG (Li-Cor, Cat# 926-32213, 1:7500 dilution), and IRDye® 680RD donkey anti-mouse IgG (Li-Cor, Cat# 926-68072, 1:7500 dilution) according to Li-Cor Western blot detection protocol (manufacturer's instructions). Membranes were scanned using the Odessey CLx Gel imaging system (Li-Cor). Uncropped scans are supplied at the end of the in the Supplementary Information file (Supplementary Fig. 21, 22).

## Cytokine analysis by multiplexed luminex or ELISA assays

Cytokine analysis in cell culture supernatants was performed using a magnetic bead-based 48-plex Luminex kit from MilliporeSigma (MIL-LIPLEX® MAP Human Cytokine/Chemokine/Growth Factor Panel A, Cat# HCYTA-60K-PX48) or 4-plex Luminex kit for CXCL10, IFNβ, IL-6, TNFα from R&D Systems (Human XL Cytokine Luminex Performance Panel, Cat# FCSTM18-04). Mouse serum cytokines were measured using the bead-based Milliplex Mouse Cytokine/Chemokine Magnetic Bead Panel (MCYTMAG-70K-PX32- Premixed 32-plex kit) from MilliporeSigma according to the manufacturer's directions. For analyte measurements, the plates were run on the FLEXMAP 3D® Luminex analyzer (Build: 4.2.1513.0) using xPONENT software, and analyte concentrations (pg/mL) were determined using Belysa Immunoassay Curve Fitting Software (MilliporeSigma, version 1.0.19). Only cytokines with a measurable increase were plotted. Human CXCL10 and IFNλ1/λ3 in culture medium were analyzed by Duo Set ELISA kits from R&D Systems (Cat# DY266 and DY1598B respectively). Dose-response curves were generated using GraphPad Prism software (version 9.3.1). $EC_{50}$ values were determined from four-parameter curve fitting.

## Gene expression analysis

Gene expression analysis of SKOV3 tumors harvested from CB.17 SCID mice was performed using NanoString. RNA was extracted from FFPE tumor tissue using the Qiagen RNeasy FFPE kit according to kit

 

instructions. For gene expression analysis of co-cultures, cells were harvested post-treatments and RNA was extracted using Qiagen RNeasy mini kit. 150 ng RNA per sample was analyzed on NanoString nCounter Max system (version 3.0.1.4) using the nCounter human PanCancer Immune Profiling code set (NanoString, XT-CSO-HIP1-12, Cat# 115000132) or mouse PanCancer Immune Profiling code set (NanoString, XT-CSO-MIP1-12, Cat# 115000142) and nCounter Standard Master Kit (NanoString, NAA-AKIT-048, Cat# 100054). Data was analyzed using the nSolver Advanced Analysis Software 4.0 (NanoString, version 2.0.115). Background thresholding was set to the geometric mean of negative controls. Both positive control and code set content (housekeeping) normalization types were selected for data normalization. Geometric mean was used to compute normalization factors (range: 0.3-3 for positive control normalization and 0.1-10 for code set content normalization). No QC flags were detected in the data sets shown in this study. The Loglinear method was used to estimate the differential expression (uses the lm function to run the Wald test). Benjamini-Yekutieli method was used for p-value adjustment. Volcano plots show each gene's -log10(p-value) and log2 fold change over vehicle treatment. To generate the heat map in Fig. 7d, normalized mRNA counts were scaled by each row (gene) using the standardization method with the below given formula and includes the genes that are upregulated or downregulated greater than 2-fold with targeted ADC-wt treatment over vehicle in both tumors.

$$\frac{Counts - Mean\ of\ all\ five\ test\ conditions\ of\ each\ gene}{Standard\ Deviation\ of\ all\ five\ test\ conditions\ of\ each\ gene}$$

### Real-Time PCR

RNA was extracted from cell harvests using the Qiagen RNeasy kit according to the manufacturer's instructions. Samples were equalized based on RNA concentration and cDNA produced using the Super-Script IV VILO Master Mix with exDNase Enzyme (Thermo Fisher, Cat# 11766050). Universal Human Reference RNA was purchased from Thermo Fisher (Cat# QS0609). Gene expression assays were set up with the TaqMan Fast Advanced Master Mix (Cat# 4444556). Probe assay IDs were: Hs00171042 (ABI) for CXCL10, Hs00174131 (ABI) for IL6, Hs00601677 (ABI) for IL29, Hs00820125 (ABI) for IL28a, hs01077958 (ABI) or qHsaCEP0054112 (BioRad) for IFNβ, qHHsa-CEP0041006 (BioRad) for IL28b, Hs04189704 (ABI) for PTPRC/CD45, and Hs00901885 (ABI) for EPCAM. Probes for housekeeping genes used were Hs03929097 (ABI) for GAPDH and Hs99999903 (ABI) for ACTIN B. Reactions were run on a Quant Studio 5 rtPCR System (ABI). QuantStudio Design and Analysis software (version 1.5.1) was used for rtPCR data analysis.

### Immunohistochemistry

HER2 and NaPi2b immunohistochemistry (IHC) were performed on formalin-fixed paraffin-embedded samples cut at 5 μm from untreated SKOV3 (HER2), and OVCAR3 (NaPi2b) xenograft models, human tumor samples (NaPi2b), or 4T1-humanHER2 tumors. Deparaffinization was completed with multiple changes of xylenes and alcohols at decreasing concentrations. Manual antigen unmasking for both antibodies was done using heat-induced epitope retrieval with an electronic pressure cooker, heated to 99 °C for 20 minutes. Slides for HER2 were retrieved in EDTA buffer at pH 9.0 (Vector Laboratories, Cat# H-3300-250), and NaPi2b slides in citrate buffer at pH 6.0 (Vector Laboratories, Cat# H-3301-250). Peroxidase blocking with Dual Endogenous Enzyme Block (Agilent Technologies, Cat# S200389) followed, then primary antibody incubation with rabbit polyclonal anti-HER2 (Agilent Technologies; Cat# A0485, final dilution 1:500) or anti-NaPi2b (comprised of a human-rabbit chimera of XMT-1535[54], the antibody component of the NaPi2b ADC, final dilution 1:3000) for 30 minutes. An HRP-labeled anti-rabbit secondary antibody (Agilent Technologies,

Cat# K400311-2) was subsequently added for another 30 minutes, followed by Liquid DAB+ (Agilent Technologies, Cat# K346811-2) chromogen for visualization. Sections were then counterstained with hematoxylin, dehydrated, cleared and cover-slipped. Appropriate control samples were stained using respective staining protocols. Images were captured using the Olympus CellSens Entry 1.17 microscope camera (Olympus Corporation, Japan).

### Cell-binding assays

Binding of HER2 and NaPi2b antibodies or ADCs to SKBR3 (HER2), HCC1954 (HER2), OVCAR3 (NaPi2b), and Kuramochi (NaPi2b) cells respectively were determined by flow cytometry. Cells were incubated with the ADCs, parental antibodies, and non-binding Control ADC for 1 hour on ice and washed with ice cold PBS. Antibody concentration range: Supplementary Fig. 3a: 200−0.012 nM, 1:4 dilution (SBKR3); 300-0.0038 nM, 1:5 dilution (OVCAR3), Supplementary Fig. 11g: 400−0.024 nM; 1:4 dilution (SKBR3 and OVCAR3). 100 nM of anti-HER2 or anti-NaPi2b antibodies were used to determine HER2 and NaPi2b expression respectively in Supplementary Fig. 10b and 10d. After staining with the secondary antibodies (Alexa Fluor 647- goat anti-human IgG, 6 μg/mL, Life Technologies, Cat# A21445) for 1 hour on ice, cells were run on a MACSQuant flow cytometer (Miltenyi Biotec). Data analysis was performed using either FlowJo (version 10.8.1) or MACSQuant analysis software (version 10,2.11.1817.19623). Dose response curves were generated using GraphPad Prism (version 9.3.1). EC$_{50}$ values were determined from four-parameter curve fitting.

### Flow cytometry analysis

Immune profiling of PBMCs and cell surface expression of human FcγRs was determined by flow cytometry analysis. ~ 50,000 cells were transferred to a U-bottom 96-well plate in 4-5 replicates, washed with PBS and stained with live/dead fixable Aqua dead cell staining dye (Molecular Probes, Cat# L34966) followed by staining with fluorophore conjugated target specific (triplicates) (1:300 dilution) or isotype control antibodies (1:300 dilution) in flow cytometry staining buffer (Invitrogen, Cat# 00-4222-26) for 30 minutes at room temperature. Cells were washed and resuspended in 1-2% PFA in flow cytometry staining buffer and run on a MACSQuant flow cytometer (Miltenyi Biotec). Data analysis was performed by FlowJo software (version 10.8.1). Gating was determined based on the isotype control antibody-stained populations. An example gating strategy is given in Supplementary Fig. 20. Antibodies were purchased from BioLegend: PerCP/Cy5.5-CD16 (Clone 3G8, Cat# 302027), PE-CD32 (Clone FUN-2, Cat#303206), APC-Cy7-CD64 (Clone 10.1, Cat# 305025), APC-CD45 (Clone 2D1, Cat#368512), FITC-CD3 (Clone UCHT1, Cat#300406) and Pacific Blue-CD14 (Clone 63D3, Cat#367122).

### Flow cytometry-based antibody/ADC internalization assay

FITC-conjugated antibodies or ADCs (concentrations are indicated in the respective figure legends) were incubated on non-coated control plates or NaPi2b recombinant antigen-coated plates in a culture medium for 15 minutes at 37 °C in an incubator. 50,000 THP1 cells were added and incubated for 5−6 more hours. THP1 cells were harvested, washed, and stained with PE-anti-FITC antibody (BioLegend, Cat# 408308, clone FIT-22, 1:300) followed by flow cytometry analysis as described above to determine the cell surface-bound (PE-high) vs internalized (PE-low) FITC-conjugated antibodies. Unstained cells were included as a control instead of isotype control antibody-stained cells.

### Cell separation / isolation

Enriched monocytes were isolated from human PBMCs using the human monocyte enrichment magnetic separation kit (with CD16 depletion) from Stem Cell Technologies (Cat#19359). Monocyte depletion from PBMCs was performed using the EasySep Human CD14 Positive Selection Kit II from Stem Cell Technologies (Cat# 17858).

Cancer cells were isolated from co-cultures by removing the immune cells using a Do-It-Yourself™ positive selection kit (StemCell Technologies, Cat# 17698) and an anti-human CD45+ (Clone HI30) antibody (StemCell Technologies, Cat# 60018) to select for CD45+ cells according to manufacturer instructions. Briefly, SKBR3-NucRed cancer cells were plated at a density of 400,000 cells per well in a 6-well plate and allowed to adhere overnight. The next day, 800,000 human PBMCs were added to the cultured cancer cells and the respective treatment was added. After 5 hours, co-cultures containing cancer cells and PBMCs were collected, washed, and incubated with the anti-human CD45+ selection cocktail for 15 minutes and then incubated with magnetic rapid spheres for 10 minutes. Samples were then placed in the EasyEights™ EasySep™ Magnet (StemCell Technologies, Cat# 18103) for 10 minutes. Supernatants were then collected to purify the CD45 negative cells and pelleted for RNA extraction.

### Biolayer interferometry

Binding of the Fc wt or mutant HER2 and NaPi2b ADCs to Fcγ receptors I and II were determined by Biolayer Interferometry assay on a ForteBio Octet QKe (Octet BLI Systems, Sartorius). All dilutions and baseline measurements were done in 1x kinetics buffer. Octet kinetics buffer (1x) was used to pre-wet anti-human Fab-CH1 2nd generation (FAB2G) biosensors (Sartorius). Baseline measurements were performed for 60 seconds prior to and post capture (300 seconds) of 1 μg/mL of each test article. Association of the test articles to recombinant FcγRI and FcγRII was then measured for 250 seconds from at least 2 concentrations diluted two-fold from 100 nM and 1000 nM recombinant protein, respectively. This was followed by a dissociation step for 600 seconds. Forte Bio analysis software (version 9.0) was used to generate plots using a 1:1 model and global fit average.

### In vivo animal studies

All in vivo studies were approved by the Institutional Animal Care and Use Committee (IACUC) of Charles River Discovery Services (CRL, North Carolina, USA; SKOV3 and 4T1-hHER2 xenograft models, study protocol numbers: 980701, 990202, 980701) or Translational Drug Development, LLC (TD2, Arizona, USA; OVCAR-3 xenograft model, study protocol numbers: 19021, 20013). Both facilities are accredited under the Association for Assessment and Accreditation of Laboratory Animal Care International (AAALAC). Sex was not considered for the purpose of the studies. Female mice were used for practicality reasons (ie to reduce the chance of fighting when housed in groups). Mice were housed in microisolator cages on a 12-hour light/dark cycle at 70-72 °F (CRL) or 68-79 °F (TD2) and 40-60% (CRL) or 30-70% (TD2) humidity. Ten-week-old female CB.17 SCID mice (Fox Chase SCID, *CB17*/Icr-*Prkdc^{SCID}*/IcrIcoCrl, Charles River Laboratories) were subcutaneously inoculated in the right flank with $1x10^7$ SKOV3 cells in 50% Matrigel®. Fourteen to fifteen-week-old female CB.17 SCID (C.B-17/IcrHsd-*prkdc^{SCID}*, Envigo) or eleven- to twelve-week-old female athymic nude (Hsd: *Athymic Nude-Foxn1^{nu}*, Envigo) mice were subcutaneously inoculated in the right flank with $5x10^6$ OVCAR-3 cells in 50% Matrigel®. Ten-week-old female BALB/c mice (*BALB/cAnNCr*l, Charles River Laboratories) were subcutaneously inoculated in the right flank with $2x10^6$ 4T1 cells engineered to overexpress human HER2 as described above (4T1-hHER2). Animals were randomized using study management software when tumors reached 60–100 mm³ (SKOV3, 4T1-hHER2; Research Flow Management System (RFMS) version 2.0.7422) or 52–247 mm³ (OVCAR-3; Study Director version 3) and treated according to the doses, schedules, and routes shown in the figures (N = 8–10 mice/group). Tumors were measured by caliper twice weekly and tumor volumes were calculated using the formula: width² × length / 2. Individual animals were euthanized when tumors reached 1000–1500 mm³, below the maximum tumor burden limits of 2000 mm³ (CRL) and 3000 mm³ (TD2) as defined by the IACUC of each facility. Kaplan-Meier plots show the percentage of animals per group with tumors <500 mm³ over the course of the study. In vivo figures were created using GraphPad Prism (version 10.1.2). For the PD (pharmacodynamic) study, 12-week-old female CB.17 SCID mice bearing SKOV3 xenografts were randomized into treatment groups (n = 10 mice/group) when tumors reached 108–172 mm³. Animals were given a single, intravenous injection of each treatment. Serum was collected at 6, 12, 24, and 72 hours following treatment (n = 5/ mice/group/timepoint). Tumors were collected at 12 and 72 h following treatment, formalin fixed, and paraffin embedded (n = 5/mice/group/timepoint).

### PK analysis

Ten-week-old female CB.17 SCID mice were dosed intravenously as a single dose with vehicle or HER2-ADC (3/0.1 mg/kg) (antibody/payload, n = 3 for each group). Blood was serially collected from all animals at 1, 24, 48, 72, 96, 168, 240, and 336 hours following treatment and immediately diluted 1:10 with acidic buffer (0.6% BSA (w/v), 5 mM EDTA in 100 mL PBS + 15.34 ml 10 mg/mL citric acid), for a total volume of 0.1 mL. Diluted whole blood was snap-frozen on dry ice and stored at −80 °C until analysis for total antibody and conjugated drug. Total antibody and conjugated drug concentrations of the ADC in the plasma samples were measured using an MSD-ECL sandwich immunoassay and immune-capture-mass spectrometry technique respectively.

### Ex vivo patient derived tumoroid culture assays

Ex vivo human tumor fragment culture assays were performed by Nilogen Oncosystems according to their company guidelines. All tissues were collected under IRB approval (Ohio State Biomedical Institutional Review Board, study protocol number: 2014H0130). Tier 1 characteristics of fresh human tumor tissue according to the BRISQ guidelines[56] are shown in Supplementary Table 1. Fresh tumors from two different ovarian cancer patients were each fragmented into tumoroids immediately and pooled to preserve tumor heterogeneity. A small fraction of each tumor prior to fragmentation was processed into FFPE for retrospective determination of the tumor subtype and NaPi2b expression by IHC. Four hundred tumoroids were dissociated into single cells and analyzed by flow cytometry for profiling, and four hundred tumoroids per well were seeded into 96 well-plates for cytokine analyses of the supernatants and NanoString gene expression analysis of the RNA extracts 8 hours and 24 hours after treatment with 20 nM (based on payload) of Fc-wt NaPi2b ADC, Fc-mutant NaPi2b ADC, free STINGa, and Control ADC. PMA and Ionomycin (PMA-I) treatment was included as a control for intact immune cell function. The histopathology of tumor #1 and tumor #2 was determined retrospectively as well-differentiated endometroid carcinoma and high-grade serous carcinoma respectively. High-content confocal images of tumoroid cultures were acquired 6 hours and 72 hours after treatments. Tumoroid cultures were first incubated with cell viability stains followed by treatment with vehicle, Alexa Fluor-488 (AF-488)-conjugated non-binding Control ADC or NaPi2b ADC (Fc-wt) (20 nM based on payload).

### Data analysis / statistics / reproducibility

All in vitro experiments were performed using 2–3 replicates (biological). The number of replicates for each experiment is shown in figure legends. Flow cytometry analyses of cell surface expression of target proteins (only in the absence of treatments) were performed using 3 independent technical replicates. Data shown are representatives of at least two independent experiments with consistent observations unless otherwise indicated in the figure legends. Multiplexed cytokine, mRNA expression, and PK analyses were performed once using samples from 3-5 different animals or biological replicates. Fresh tumor fragment culture assays were performed once using samples from two different tumors. Means, standard deviation (SD), and standard error of mean (SEM) were calculated using the GraphPad Prism Software (version 9.3.1).

## Reporting summary

Further information on research design is available in the Nature Portfolio Reporting Summary linked to this article.

## Data availability

Data are available within the Article, Supplementary Information, or Source Data file. Source data are provided in this paper.

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

## Acknowledgements

The authors wish to thank Charles River Laboratories and TD2 for their contributions to the in vivo work; Nilogen Oncosystems for performing the ex vivo tumor fragment studies using their fresh human tumor 3D platform; Winnie Lee and Liu Liang Qin for technical help with preliminary experiments; Keith Bentley, Joshua D. Thomas, Eoin Kelleher and Susan Clardy for contributing to the synthesis and analytical characterization of the reagents, Ling Xu and Elena Ter-Ovanesyan for PK analysis. Cartoons in Fig. 1 (A, B, C, D, F), 6-G, 7-A, and supplementary figs. 2, 4-A, 6-A, 13-H were created with BioRender.com, released under a Creative Commons Attribution-NonCommercial-NoDerivs 4.0 International License.

## Author contributions

Conceptualization, writing/revision of the manuscript: N.M.C., M.Damelin. Study design, data acquisition, analysis, interpretation: N.M.C., T.M., J.S.J., P.S., K.C.C., K.L., M.Dolan, R.M., C.R., and C.C. Supervision: N.M.C. and MDamelin, T.L., D.T., J.D., and R.B. All authors read and approved the manuscript.

## Competing interests

All authors are current or former employees of Mersana Therapeutics and may hold stock options and/or equity in Mersana Therapeutics Inc. Related patents: US11,155,567: Sting agonist compounds and methods of use, WO2021/202984: Antibody-drug conjugates comprising sting agonizts.
