## [Peer Review File · Nature Communications]

Tumor Cell-Directed STING Agonist Antibody-Drug Conjugates Induce Type III Interferons and Anti-Tumor Innate Immune ResponsesREVIEWERS' COMMENTS:

Reviewer #1 (Remarks to the Author):

Dear editor,

The work by Cetinbas et al is a proof of concept study, in which authors analyzed the therapeutic potential of STINGa ADCs using as antibody moieties two humanized mAbs directed against two well-known internalizing antigens, HER2 receptor and NaPi2b which are validated target for ADC therapy in different cancers (such as breast, gastric and ovarian).

The topics covered in the article are of extreme relevance being nowadays the interest in the development of novel ADCs in the field of targeted therapy in cancer is constantly increasing. Moreover, here authors explore the potential of a novel type of ADC in which classical cytotoxic payloads have been replaced by antitumor immune stimulator (STING agonist), which make even more intriguing this therapeutic approach for the scientific community.

The study is generally well conducted and the amount of data presented is of remarkable quantity; the majority of the experiments appears to be well designed with appropriate controls. The flow of the manuscript is not always smooth and is sometimes difficult to follow. It could be improved by rearranging some panels and/or deleting some data that could appear redundant.

The hypothesis behind this strategy is that the delivery of STING agonist mediated by the antibody is able to efficiently promote a potent antitumor immune response mediated by STING pathways, which are activated both in cancer and myeloid cells. Mechanistically, the fascinating novelty proposed by the authors for this type of STINGa-ADC relies in the fact that the ADC, in the presence of tumoral antigen, can be internalized both by cancer and myeloid cells.

Overall this is an important piece of work with potential interest for Nat Comm readers however there are some important issues to be clarified by the authors before this manuscript can be accepted for publication:

Major points:

- I found really weird and disappointing that the recent work by Wu, You-Tong et al. "Tumor-

targeted delivery of a STING agonist improves cancer immunotherapy.” Proceedings of the National Academy of Sciences of the United States of America vol. 119,49 (2022): e2214278119. doi:10.1073/pnas.2214278119 ADC-STINGa , is not cited and discussed by the authors. This is the first report in which is described a STING/ADC where the EGFR has been used as tumor antigen. The authors are invited to define novelty/differences, although in an indirect comparison, between their ADCs and the one recently described in the Wu, You-Tong et al paper.

-ADC/STINGa development: as for ADCs generation, the authors refer to their granted patents, however it would be more accessible for the readers having here in this manuscript in the method section detailed information describing how these novel ADC were obtained. As example, It's not clear whether STING agonists have been couple through random or site specific conjugation. No information are reported on the DAR value for each ADC. No quality controls about the obtained compounds are shown.

-More importantly, no information are provided about Pharmacokinetic (PK) and Pharmacodynamic (PD) Properties of these novel STING/ADCs.

-Authors clearly demonstrate that STINGa/ADCs activate STING pathway in myeloid cells in an antigen and Fc-dependent manner. Moreover, they show that the ADCs are internalized both by myeloid cells (through FcγRI) and tumor cells (through the tumor antigen). Indeed, it's well described by the authors the ability of STING/ADC to activate STING response both in tumor and myeloid cells. However, it's not clear from a mechanistic and quantitative point of view how this intriguing processes could be regulated. How the process of concomitant internalization can take place and in what proportion? Is it regulated by affinity , i.e. antigen/antibody affinity or FcγRI/Fc binding affinity? Could the different expression of tumor antigen modify the ADC/STING internalization rate between tumor and myeloid cells? In line with this confocal imaging in a co-culture system could better clarify the amount and the kinetic of internalized ADC into tumor cells in presence of myeloid cells. Indeed, in a co-culture system (on the contrary of what can be observed using an antigen-coated plate) a competition of the two internalization process may take place.

- The mechanism described by the authors for the internalization of ADCs on myeloid cells should be shared by other classical ADCs, in view of the fact that the internalization process is strictly driven by the FcγRI/Fc interaction. For example, can the internalization data presented in Figure 1H be obtained using bare or DM-1/DXd conjugated trastuzumab?

Furthermore, in Figure 3c the authors show a multiplex cytokine analysis where both the wt and the Fc ADC mutant led to a significant increase of several cytokines in the PBMC/Skbr-3 co-culture system. Since the monocyte/ADC interaction is driven by FcγRI/Fc binding, it would be essential here to demonstrate the activation of the STING-dependent cytokine profile by comparing the cytokine induction of STINGa/ADC with other classical ADCs loaded with cytotoxic compounds, such as T-DM1 or T-DXd.

- Figure 3A and 3B show the in vivo antitumor activity of the two STING/ADCs. As illustrated, there is also significant activity for Fc-mutated ADCs, however according to the graphs it appears that while the anti-tumor response of STING/ADC is stable and long-lasting, Fc-mutated treated tumors start to regrow after 35-42 days from the start of treatment. What happens next? Do the authors have data after 45 days? It is possible to hypothesize that the wt-STING/ADC response would be longer and more stable than Fc-mutant... Kaplan-Meier survival analysis could be added if data are available at longer time points.

- Overall, a significant weakness of this paper is that the evaluation of the therapeutic activity of these new STING/ADCs is based on two animal studies in which the commercial cell lines Skbr-3 and OVCAR3 were used as a subcutaneous tumor model. . Further data would be essential to define the real potential antitumor activity of these new therapies: preclinical PDX models with different tumor antigen expression levels and orthotopic or pseudometastatic models would be informative to fully understand the potential of these new ADCs. Furthermore, comparison of the therapeutic activity of these new ADCs with approved ADCs, such as T-DM1 or T-DXd for HER2 antigen, would strengthen their findings.

- The authors used SCID immunodeficient mice for in vivo assays. Why did the authors not use immunocompetent animals, i.e. syngeneic models? It would be of great interest to analyze the STING/ADC response in the context of a complete immune response and possibly in combination with approved immune checkpoint inhibitors.

-Figure 4

Panel A/B : it's not clear here in the co-culture system where the RNA analyzed come from. Why in the expression analysis as control is used the control ADC and for ELISA both Fc-mutant and control ADC. It's quite confounding that different type of controls are used during similar assays along the paper. More importantly, is the induction of such type III IFNs specific for STING/ADC ? Again, T-DM1 or T-DXd should be used here as control to

demonstrate STINGa dependence on type III IFNs induction.

-Figure 4 E/F : authors should clarify the reason why there's IFN λ induction in HCC1594 triple negative monocultures and not in HER2+++ Skber3 cells.

-Figure 5A: control on cell fractionation is missing. How authors can exclude a cross contamination of the two cell populations? In addition, results, in terms of mRNA induction are not consistent with the Figure 4B: indeed, as example, for IFN λ 1 >600 fold induction is observed upon 20nM (payload) stimulation while in figure 4B <200 fold induction is observed in response to 50nM stimulation. Overall, data presented in Figure 4B appear to be redundant.

-An important point raised by the authors is that according to the data presented it's reasonable to state that functional tumor STING pathway is necessary for type III IFN induction in immune co-cultures system, even in the presence of an intact STING signalling in immune cells. To support authors conclusions it would be of importance to evaluate whether type III IFN induction by STINGa/ADC correlate with cancer STING expression in a panel of antigen positive (HER2 or NaPi2b) tumor cells.

- some of figure panels cannot be reviewed due to poor resolution. High quality images should be provided by the authors for a proper evaluation.

Minor issues:

All the method section should be implemented as indicated below:

-no information are provided in the method section about the cell lines used in the study

-no methodological information are described relative to the co-culture system.

-no information are provided about the fractionation methods (Fig 5A)

-no details are present in the method section about western blotting and immunofluorescence.

-no information are provide about cytokine analysis on serum samples (Figure 2C).

-flow cytometry analysis section should be implemented providing all the information about all the assays described in the paper (Figure 1i, Figure 7C, Figure S4A, Figure S6C).

Reviewer #2 (Remarks to the Author):

In this manuscript, Malli Cetinbas et al examine the potential of tumor cell directed STING agonist ADCs, demonstrating internalization by myeloid cells and efficacy in xenografts in SCID mice. They also unveil a role for tumor cell STING induction by myeloid CM that amplifies tumor cell sensitivity, and a novel function of type III IFNs in promoting activity. They also use short term ovarian cancer explant cultures to validate results in human tissue.

Overall, this is a nice manuscript developing a novel agent with quite a bit of data. However, there are several issues that need to be addressed. Figure quality/resolution is also a bit low.

1. Introduction – The paragraph on the role of tumor cell STING is confusing. Silencing or suppression of STING is not contradictory with the fact that others have shown that cancer cell STING is required for response to radiation or other DNA damaging agents. In fact, it further reinforces the importance of tumor cell STING – for example Kitajima et al (Cancer Discov 2019) have shown that silencing of STING is linked with immune evasion in KRAS-LKB1 mutant lung cancer, with enhanced susceptibility to DNA damage induced micronuclei when restored (Cancer Cell 2022). Similarly, Falahat et al, (Nature Comm 2023) have shown that murine syngeneic melanoma lines silence STING, and when it is restored epigenetically this also unveils the key role of tumor cell STING.

2. Figure 2 –The in vivo data are nice, but what is the mechanism for durable anti-tumor immunity? Do they observe NK cell infiltration into treated tumors in SCID mice, which retain NK cells? It is important to understand if this is purely tumor cell/macrophage intrinsic or if NK cells are involved in vivo. Can they eradicate larger tumors? Treatment is started at a very small size. Is the effect lost in NSG mice which lack NK cells?

3. Figures 3-6 – Again a comprehensive set of convincing data are presented. However, results are again mechanistically under-explored with respect to their simple model that only myeloid cells alone are involved. The key experiment presented in Fig 3D and E is extremely confusing with respect to how it is labeled, as I am not even sure what to compare since the color for the PBMC co-cultures vs monocyte co-cultures are the exact

same. If the curves are really that overlapping between 2 independent experiments I would be very surprised. I also can't find experimental details regarding how they purified monocytes, and they only show a reduction in T cells and not NK cells in Fig S3. Another simple experiment would be to test if B2M knockout in cancer cells impairs response? This experiment would help to determine if PBMC T cell alloreactivity is involved, vs NK cell activity, since one would predict loss of killing with B2M KO if there is a role for T cells, or increased killing if NK cells. Or no effect if this is purely related to macrophage activation and direct tumoricidal activity as their model suggests.

Reviewer #3 (Remarks to the Author):

This study aims to test an ADC that targets HER2 or SLC34A2 to deliver STING agonists as a payload. The study team showed that ADC was internalized to myeloid cells in a FcR-I and tumor antigen-dependent fashion. This ADC induces type III interferon, which contributes to an anti-tumor immune response. The main conclusion is that a previously underappreciated role of type III IFN in anti-tumor activity can be elicited by STING-inducing ADC. Although the system appears robust in cytokine production, the study lacks the rigor to generate sufficient impact and conceptual innovation. The study is heavily dependent on in vitro co-culture and SCID mice. STING-mediated immunity depends on CD8 T cells. Syngeneic models and GEMMs are more appropriate to advance this line of agents conceptually. The toxicity study is very limited. Type III IFN shares many signaling components with type I IFN but target mucosal surfaces. It is not surprising that this product induces type III as well, but the significance of type III-specific target tissue is not addressed. Without the use of syngeneic models that utilize IFNLR deficient tumors or IFNLR-deficient hosts, it is not possible to definitely reach the central conclusion that type III contributes to anti-tumor immunity in vivo. In addition, there are specific comments below.

1. The heavy use of SCID mice dampens the rigor of the entire study. For example, Figure 2 concludes that ADC is more effective than diABZI. It is known that STING-mediated immunity depends on T cells. Although ADC might be more effective in this setting, a fair comparison would be in immunocompetent mice.
2. The toxicity studies are quite rudimentary. Mice are more tolerant to immune-related toxicities. No change in body weight does not mean much. In fact, the worrisome clinical

toxicity performance (the first case who was treated with ADC passed away) suggests that a more comprehensive toxicity study in large animals, including NHP, is needed. It is possible that the patient died of other reasons. However, the study presented here lack key elements for a clinical-grade new therapeutic agent.

3. Tumor antigen expression has high variability. The ADC effectiveness appears to be dependent on tumor antigen expression. It is unclear how effective this treatment would be for tumors with heterogenous antigen expression levels. The single-antigen design further dampens its clinical potential.

4. Many cancer cells are defective in STING signaling. The conclusion that cancer cell STING is required for type III IFN production needs many more cell lines than a single cell line.

5. Type III IFN and type I IFN share many signaling components. It is not surprising that this particular ADC can induce type I and type III IFNs. A more important question is how therapeutically significant type III IFN is in this setting. There is no attempt to address whether the deficiency in cancer cell specific or host specific type III IFN receptor would make a difference.

6. Type III IFN has a different target tissue, which is usually mucosa. The contribution of type III signaling-specific target tissue in ADC-mediated anti-tumor immune response is not addressed.

7. Cancer cells are defective for STING signaling for many reasons. The exclusive focus on STING expression level is narrow in scope and not novel.

8. The impact of ADC on T cells in the TME and draining lymph nodes are largely ignored in this study.

Summary of Responses to Reviewer Comments:

We thank the Reviewers for their thorough review and helpful comments. Our revised manuscript has addressed these comments, as detailed below, and as a result is greatly improved. As explained below, a few specific suggestions for experiments were sensible in principle yet not feasible from a technical standpoint. We are pleased that overall the Reviewers found the work novel and relevant, and appreciated that the findings are broadly applicable to the field's intensive exploration of STING agonism to induce anti-tumor immunity.

Summary of Improvements in Revised Manuscript:

- 10 new figures with multiple panels added to Supplementary Information
- 4 figures provided for reviewer only (to be published with XMT-2056 manuscript)
- Minor revisions to the original figures
- 3 references added
- Additions/clarifications throughout the manuscript text

Please find below the responses to specific Reviewer comments.

Reviewer #1 (Remarks to the Author):

Dear editor,

The work by Cetinbas et al is a proof of concept study, in which authors analyzed the therapeutic potential of STINGa ADCs using as antibody moieties two humanized mAbs directed against two well-known internalizing antigens, HER2 receptor and NaPi2b which are validated target for ADC therapy in different cancers (such as breast, gastric and ovarian).

The topics covered in the article are of extreme relevance being nowadays the interest in the development of novel ADCs in the field of targeted therapy in cancer is constantly increasing. Moreover, here authors explore the potential of a novel type of ADC in which classical cytotoxic payloads have been replaced by antitumor immune stimulator (STING agonist), which make even more intriguing this therapeutic approach for the scientific community.

The study is generally well conducted and the amount of data presented is of remarkable quantity; the majority of the experiments appears to be well designed with appropriate controls. The flow of the manuscript is not always smooth and is sometimes difficult to follow. It could be improved by rearranging some panels and/or deleting some data that could appear redundant.

The hypothesis behind this strategy is that the delivery of STING agonist mediated by the antibody is able to efficiently promote a potent antitumor immune response mediated by STING pathways, which are activated both in cancer and myeloid cells.

Mechanistically, the fascinating novelty proposed by the authors for this type of STINGa-ADC relies in the fact that the ADC, in the presence of tumoral antigen, can be internalized both by cancer and myeloid cells.

Overall this is an important piece of work with potential interest for Nat Comm readers however there are some important issues to be clarified by the authors before this manuscript can be accepted for publication:

Major points:

1. I found really weird and disappointing that the recent work by Wu, You-Tong et al. "Tumor-targeted delivery of a STING agonist improves cancer immunotherapy." Proceedings of the National Academy of Sciences of the United States of America vol. 119,49 (2022): e2214278119. doi:10.1073/pnas.2214278119 ADC-STINGa , is not cited and discussed by the authors. This is the first report in which is described a STING/ADC where the EGFR has been used as tumor antigen. The authors are invited to define novelty/differences, although in an indirect comparison, between their ADCs and the one recently described in the Wu, You-Tong et al paper.

- Thank you for pointing out this oversight. This paper was published just as our manuscript was being finalized. We properly cited it in the revised manuscript and emphasized the novelty of our studies (page 25, last paragraph, citation#48).

2. ADC/STINGa development: as for ADCs generation, the authors refer to their granted patents, however it would be more accessible for the readers having here in this manuscript in the method section detailed information describing how these novel ADC were obtained. As example, It's not clear weather STING agonists have been couple trough random or site specific conjugation. No information are reported on the DAR value for each ADC. No quality controls about the obtained compounds are shown.

- We added detailed information in the revised manuscript methods section (pages 29-30) and included analytical QC results for the ADCs in the new Supplementary Fig. S1C, S1D.

3. More importantly, no information are provided about Pharmacokinetic (PK) and Pharmacodynamic (PD) Properties of these novel STING/ADCs.

- The PD data was provided in the original main figure 2C, 2D (i.e. serum cytokine and tumor gene induction profiles, Fig. 2C, Fig. 2D, Fig. S6A, S6B in the revised manuscript). In the revised manuscript, we added PK data for the same Her2-targeted ADC (Results section Page 8, lines 20-23, new Supplementary Fig. S6C). This data demonstrates an important characteristic of the ADC: its high stability in circulation, as indicated by parallel curves of total antibody and conjugated drug.

4. Authors clearly demonstrate that STINGa/ADCs activate STING pathway in myeloid cells in an antigen and Fc-dependent manner. Moreover, they show that the ADCs are internalized both by myeloid cells (trough FcγRI) and tumor cells (trough the tumor antigen). Indeed, it's well described by the authors the ability of STING/ADC to activate

STING response both in tumor and myeloid cells. However, it's not clear from a mechanistic and quantitative point of view how this intriguing processes could be regulated. How the process of concomitant internalization can take place and in what proportion? Is it regulated by affinity , i.e. antigen/antibody affinity or FcγRI/Fc binding affinity? Could the different expression of tumor antigen modify the ADC/STING internalization rate between tumor and myeloid cells? In line with this confocal imaging in a co-culture system could better clarify the amount and the kinetic of internalized ADC into tumor cells in presence of myeloid cells. Indeed, in a co-culture system (on the contrary of what can be observed using a antigen-coated plate) a competition of the two internalization process may take place.

- The reviewer raises important questions about the kinetics/dynamics of ADC internalization into tumor cells vs myeloid cells with respect to antigen expression levels, binding affinity to antigen vs FcγR, as well as the target antigen vs FcγR cell surface recycling kinetics. We have been exploring these questions, which require extensive work using multiple models and assay systems. To address the specific question about ADC or antibody internalization into myeloid cells in the presence of cancer cells, we included data in the original Fig. 1G demonstrating efficient activation of THP1 myeloid cells by both HER2 ADC-wt Fc and Napi2b ADC-wt Fc in SKBR3 (HER2) and OVCAR3 (Napi2b) cancer cell co-cultures in an antigen and Fc-dependent manner. We have added more data in the revised manuscript demonstrating that the tumor cell-targeted mAb is internalized efficiently into myeloid cells in cancer cell co-cultures (Results section, Page 6, end of 2nd paragraph, new Fig. S4B). Due to the extensive work needed to answer these questions, we will need to publish an in-depth study in a separate manuscript. We have added a brief paragraph to the Discussion (Page 25, 1st paragraph) mentioning that further studies are needed to elucidate the details of kinetics/dynamics of ADC internalization into cancer cells vs myeloid cells with respect to target expression levels, ADC concentration, receptor binding / recycling kinetics.

5. The mechanism described by the authors for the internalization of ADCs on myeloid cells should be shared by other classical ADCs, in view of the fact that the internalization process is strictly driven by the FcγRI/Fc interaction. For example, can the internalization data presented in Figure 1H be obtained using bare or DM-1/DXd conjugated trastuzumab? Furthermore, in Figure 3c the authors show a multiplex cytokine analysis where both the wt and the Fc ADC mutant led to a significant increase of several cytokines in the PBMC/Skbr-3 co-culture system. Since the monocyte/ADC interaction is driven by FcγRI/Fc binding, it would be essential here to demonstrate the activation of the STING-dependent cytokine profile by comparing the cytokine induction of STINGα/ADC with other classical ADCs loaded with cytotoxic compounds, such as T-DM1 or T-DXd.

- We have data demonstrating that the fluorophore-conjugated parental mAb without STING agonist payload can similarly internalize into myeloid cells in the presence of the target antigen-expressing cancer cells (new Fig. S4B),

suggesting that any wild type human IgG1 antibody and ADC (including the cytotoxic ADCs T-DM1 or T-DXd) can be internalized into myeloid cells upon target antigen binding, unless the payload and conjugation method interferes with the process.

- Re- Fig 3c multiplexed cytokine assay: The targeted ADCs with wt Fc and mutant Fc both induce significant levels of cytokines, with wt ADC inducing a stronger effect due to the ability to deliver the STINGa into myeloid cells in addition to cancer cells. Lack of cytokine induction with the unconjugated Trastuzumab in this assay shows that the cytokine induction is dependent on the STING agonist. Moreover, the data we provide for the reviewer here (Figures for Reviewer #1-Q5) shows that the T-DXd-mediated cancer cell death is not enhanced in PBMC co-cultures, suggesting that there isn't significant immune-contribution to the T-DXd anti-tumor activity in this setting.

Reviewer #1 Q5:

For reviewer only - will not be included in the manuscript

- HER2-STINGa-ADC (XMT-2056) activity is enhanced in PBMC co-cultures, whereas no immune cell contribution is seen with ENHERTU (T-DXd) (demonstrating differentiating MOA)

6. Figure 3A and 3B show the *in vivo* antitumor activity of the two STING/ADCs. As illustrated, there is also significant activity for Fc-mutated ADCs, however according to the graphs it appears that while the anti-tumor response of STING/ADC is stable and long-lasting, Fc-mutated treated tumors start to regrow after 35-42 days from the start of treatment. What happens next? Do the authors have data after 45 days? It is possible to hypothesize that the wt-STING/ADC response would be longer and more stable than Fc-mutant... Kaplan-Meier survival analysis could be added if data are available at longer time points.

- The Fc-mutant ADCs activate STING in cancer cells only, while the wt Fc ADCs activate STING in both cancer cells and myeloid cells, which results in a stronger response. Consequently, the wt ADC has greater activity than the Fc mutant ADC in the *in vitro* assays and the *in vivo* tumor studies. In the revised

manuscript, to highlight this point further, we provide the tumor growth data for individual animals as well as the Kaplan-Meyer survival plots (See new Supplementary Fig. S8A-S8D).

7. Overall, a significant weakness of this paper is that the evaluation of the therapeutic activity of these new STING/ADCs is based on two animal studies in which the commercial cell lines Skbr-3 and OVCAR3 were used as a subcutaneous tumor model. Further data would be essential to define the real potential antitumor activity of these new therapies: preclinical PDX models with different tumor antigen expression levels and orthotopic or pseudo metastatic models would be informative to fully understand the potential of these new ADCs. Furthermore, comparison of the therapeutic activity of these new ADCs with approved ADCs, such as T-DM1 or T-DXd for HER2 antigen, would strengthen their findings.

- We agree that in the case of a manuscript that introduces a clinical compound, pharmacology studies in a broader range of tumor models would be informative. However, this study has focused on elucidating the mechanism of action of tumor cell-targeted STINGa-ADCs. To ensure that the results were generalizable, we performed studies with 2 tumor antigens and their associated tumor models in vitro and in vivo in different tumor types. Additionally, the original manuscript included cytokine induction and tumor cell killing data in fresh human ovarian tumor cultures ex vivo (Fig. 7). In the revised manuscript we added data in additional cancer cell-PBMC co-cultures with HCC1954 (high HER2) and Kuramochi (low NaPi2b) (See new Supplementary Fig. S10), as well as in vivo efficacy data in a syngeneic tumor model (See new supplementary Fig. S7). The mechanism of action is consistent across the variety of tumor models, tumor types, and tumor antigens that we have evaluated. Of course, as predicted by clinical experience in oncology, one compound or even one MOA will not be effective against all tumors, and resistance mechanisms are likely to exist or arise; these topics will be the source of continued studies with STINGa-ADCs.

8. The authors used SCID immunodeficient mice for in vivo assays. Why did the authors not use immunocompetent animals, i.e. syngeneic models? It would be of great interest to analyze the STING/ADC response in the context of a complete immune response and possibly in combination with approved immune checkpoint inhibitors.

- Syngeneic mouse models are required in order to study the adaptive immune responses to STING agonism and there are several studies published on this topic. This study explores the anti-tumor activity directly mediated by the innate immune responses induced by STING, which has been overlooked in the field. SCID mice are suitable for such studies since they retain an intact myeloid immune compartment. The advantage of using SCID mice is that they accommodate human tumor xenografts, such that candidate therapeutics can be evaluated. We have obtained similar efficacy results in the 4T1-human

HER2 engineered syngeneic tumor model in Balb/c, which we have added to the revised manuscript (new Supplementary Fig. S7). We also show below additional work in a syngeneic model with surrogate of our clinical candidate XMT-2056 (HER2-targeted STINGa-ADC) including in combination with anti-PD1; these results will be published separately with the XMT-2056 manuscript and are provided here for the reviewers only (Figures for Reviewer #1-Q8b). We have conducted other studies with STINGa-ADCs in Balb/C and C57BL6 strains with similar results.

Reviewer #1 Q8b:

For reviewer only - will not be included in the manuscript

- Efficacy study in the EMT6-ratHER2 – Balb/C tumor model. An XMT-2056 surrogate ADC targeting ratHER2 was used. A single IV dose of XMT-2056 surrogate ADC treatment led to complete tumor regressions at 1 mg/kg (by antibody).

- Combination of XMT-2056 surrogate ADC (0.3 mg/kg by antibody) and anti-PD-1 confers benefit in the EMT6-ratHER2 – Balb/C tumor model.

9. Figure 4 Panel A/B : it's not clear here in the co-culture system where the RNA analyzed come from. Why in the expression analysis as control is used the control ADC and for ELISA both Fc-mutant and control ADC. It's quite confounding that different type of controls are used during similar assays along the paper. More importantly, is the induction of such type III IFNs specific for STING/ADC ? Again, T-DM1 or T-Dxd should be used here as control to demonstrate STINGa dependence on type III IFNs induction.

- RNA is extracted from the co-cultures including cancer cells and PBMCs.
- We have included the Fc mutant ADC in the expression analysis and consistent with the ELISA data it induced STING pathway genes. The data is consistent throughout and we now provide the differential expression volcano plot for the Fc mutant HER2 ADC treatment over vehicle treatment, as well as bar graphs of several STING pathway genes for all treatment conditions, and the heat map of the HLA genes demonstrating robust gene induction by the targeted ADCs consistent with the rest of our results (See new Supplementary Figure S14A-S14C). We have not included this data in the original manuscript due to space/word limit considerations, but in response to this comment we have added it to the revised version.
- The lack of activity of the unconjugated trastuzumab demonstrated the requirement of the STING agonist payload for the innate immune responses mediated by the ADCs. These innate immune responses are consistent with the STING agonist payload-mediated responses. Moreover, as demonstrated in Fig. 5B-5E, type III IFN induction is dependent on the tumor intrinsic STING, together supporting the STINGa payload -mediated induction of type III IFNs by the STINGa-ADCs. Cytotoxic ADCs in general do not induce cytokine responses, however, if they do so this would be via other mechanisms, which is beyond the scope of our work.

10. Figure 4 E/F : authors should clarify the reason why there's IFNlambda induction in HCC1594 triple negative monocultures and not in HER2+++ Skber3 cells.

- HCC1954 is not a triple negative breast cancer cell line; these cells express very high levels of HER2 (See new Supplementary Fig. S15C). HCC1954 but not SKBR3 cell monocultures induce IFNlambda because – unlike most cancer cells – HCC1954 cells can activate STING in monocultures, whereas SKBR3 cells cannot. This observation was included in the original manuscript in the Results section, “Cues from primary human immune cells potentiate STING pathway activation in cancer cell monocultures” with data in the original Supplementary Fig. S8A (new Supplementary Fig. S13A). In the context of this study, HCC1954 affords a way to study STING activation in cancer cell monocultures with a Her2-targeted STINGa ADC.

-Figure 5A: control on cell fractionation is missing. How authors can exclude a cross contamination of the two cell populations? In addition, results, in terms of mRNA induction are not consistent with the Figure 4B: indeed, as example, for IFNlambda1 >600 fold induction is observed upon 20nM (payload) stimulation while in figure 4B <200 fold

induction is observed in response to 50nM stimulation. Overall, data presented in Figure 4B appear to be redundant.

- Re: cross-contamination; this is an important control and we now added new figures to the Supplementary Information demonstrating the purity of cancer cell fraction. We used a magnetic separation kit to remove the CD45+ immune cells and obtained the cancer cell fraction (CD45- supernatant). We then performed qPCR analysis using CD45 (pan-immune cell marker) and EPCAM (epithelial cancer cell marker) primers to assess the purity of cancer cell fraction. Data shows that the cancer cell fraction is free of immune cells (CD45 mRNA is not detected) (see new Supplementary Fig. S16B). This data demonstrates the robust induction of both type I and type III IFN in cancer cells in response to STING agonism, which is a novel and meaningful finding with important implications.
- The data shown in figure 5A and figure 4B are from independent experiments, and ~3-fold difference in the IFNL1 mRNA induction between different experiments is acceptable for assay-to-assay variability.
- The goal of Figure 4B is to show the induction of all three IFNL isoforms. Because the NanoString gene expression code set did not include a probe for IFNL3, we used RT-qPCR to compare all three isoforms. Whereas the goal of the Figure 5A is to compare IFNL1 vs IFN β induction by the targeted ADC in cancer cells.

-An important point raised by the authors is that according to the data presented it's reasonable to state that functional tumor STING pathway is necessary for type III IFN induction in immune co-cultures system, even in the presence of an intact STING signalling in immune cells. To support authors conclusions it would be of importance to evaluate whether type III IFN induction by STING α /ADC correlate with cancer STING expression in a panel of antigen positive (HER2 or NaPi2b) tumor cells.

- The reviewer raises a reasonable question whether there is a correlation between type III IFN induction and tumor intrinsic STING protein expression. However, in our experience there is in general no good correlation between STING protein levels and STING activation in cancer cells. Supplementary Figure S13B demonstrates this point. As seen in the western blot analysis of SKBR3 and OVCAR3 cells, the basal STING protein levels are very low, despite strong induction of type III IFNs upon STING agonism in co-cultures with PBMCs (Fig. 4B, 4E, new Supplementary Fig. S14A-S14C, Fig. S15A, S15B) suggesting that the tumor intrinsic STING protein expression at baseline do not correlate with responses to STING agonism.
- some of figure panels cannot be reviewed due to poor resolution. High quality images should be provided by the authors for a proper evaluation.

- From our side, the figures seem high resolution in the final PDF of the manuscript. We wonder if this is related to the manuscript submission system? If the reviewer can point out which figure panels need better resolution, we are happy to provide them.

Minor issues:

All the method section should be implemented as indicated below:

-no information are provided in the method section about the cell lines used in the study

- We provided the cell line information under the Methods - Cell Culture section in the original manuscript. We added specific details in the revised version.

-no methodological information are described relative to the co-culture system.

- In the original manuscript, we described the co-culture methodology under different assays used: for example the THP1/cancer cell co-cultures, and PBMC co-cultures are described in detail under the corresponding sections (Pages 30-31 in the revised manuscript).

-no information are provided about the fractionation methods (Fig 5A)

- This was inadvertently left out from methods, thank you for pointing that out. We have added it to the revised version in the Methods section – Cell Separation / Isolation, Page 36.

-no details are present in the method section about western blotting and immunofluorescence.

- In the original manuscript we provided the information about the antibodies used in the western blot analysis of the STING WT and KO cells under the corresponding methods section; in the revised manuscript we have added a separate section for the Western blot analysis and antibody information (Page 32). Immunofluorescence was described under the Ex Vivo Tumoroid Culture Assays (Page 38).

-no information are provide about cytokine analysis on serum samples (Figure 2C).

- This was inadvertently left out from methods, thank you for pointing that out. We have added it to the revised version under the cytokine analysis section (Page 33).

-flow cytometry analysis section should be implemented providing all the information about all the assays described in the paper (Figure 1i, Figure 7C, Figure S4A, Figure S6C).

- We included a general Flow Cytometry Analysis section in the original manuscript; we have now added detailed information in the Methods section for flow cytometry analyses requested by the Reviewer (Page 35).

Reviewer #2 (Remarks to the Author):

In this manuscript, Malli Cetinbas et al examine the potential of tumor cell directed STING agonist ADCs, demonstrating internalization by myeloid cells and efficacy in xenografts in SCID mice. They also unveil a role for tumor cell STING induction by myeloid CM that amplifies tumor cell sensitivity, and a novel function of type III IFNs in promoting activity. They also use short term ovarian cancer explant cultures to validate results in human tissue.

Overall, this is a nice manuscript developing a novel agent with quite a bit of data. However, there are several issues that need to be addressed. Figure quality/resolution is also a bit low.

1. Introduction – The paragraph on the role of tumor cell STING is confusing. Silencing or suppression of STING is not contradictory with the fact that others have shown that cancer cell STING is required for response to radiation or other DNA damaging agents. In fact, it further reinforces the importance of tumor cell STING – for example Kitajima et al (Cancer Discov 2019) have shown that silencing of STING is linked with immune evasion in KRAS-LKB1 mutant lung cancer, with enhanced susceptibility to DNA damage induced micronuclei when restored (Cancer Cell 2022). Similarly, Falahat et al, (Nature Comm 2023) have shown that murine syngeneic melanoma lines silence STING, and when it is restored epigenetically this also unveils the key role of tumor cell STING.

- We agree with the reviewer. We have rephrased this paragraph to be more clear.

2. Figure 2 –The in vivo data are nice, but what is the mechanism for durable anti-tumor immunity? Do they observe NK cell infiltration into treated tumors in SCID mice, which retain NK cells? It is important to understand if this is purely tumor cell/macrophage intrinsic or if NK cells are involved in vivo. Can they eradicate larger tumors? Treatment is started at a very small size. Is the effect lost in NSG mice which lack NK cells?

- Recent work by other labs (Knelson et al. doi: 10.1158/2326-6066, Li et al. doi:10.1038/s41586-022-05254-3) indicated that the NK cells play an important role in the anti-tumor activity elicited by STING agonism in the TME. We observed upregulation of NK cell markers specifically in the targeted ADC treated tumors (see below Figures for Reviewer#2-Q2).

Reviewer #2 Q2:

For reviewer only - will not be included in the manuscript

Bar graphs showing the normalized mRNA expression of mouse (A) STING pathway genes and (B) NK cell activation genes in the SKOV3 tumors treated with the indicated test articles as in Fig. 2. Data points shown in the graphs are mean of 2-3 replicates \pm SD.

Interestingly, at the time points examined, the extent of upregulation of these NK cell activation markers is less pronounced than that of the STING pathway activation markers (innate immune response markers); combined with our in vitro co-culture results (see response to Reviewer #2 – Q3) we believe that the complete tumor eradication in the SKOV3/SCID tumor model is most likely driven by strong innate immune responses induced by tumor cell-targeted ADC-mediated STING activation in cancer cells and myeloid cells, directly resulting in cancer cell death (direct tumoricidal activity as the Reviewer suggests). We do not rule out contribution of the NK cells or T cells to the anti-tumor activity and added discussion on this point (page 25-26).

- Re: Tumor size: These tumors were established before the animals were treated, and the tumor sizes were typical for immune therapy studies.

3. Figures 3-6 – Again a comprehensive set of convincing data are presented. However, results are again mechanistically under-explored with respect to their simple model that only myeloid cells alone are involved. The key experiment presented in Fig 3D and E is extremely confusing with respect to how it is labeled, as I am not even sure what to compare since the color for the PBMC co-cultures vs monocyte co-cultures are the exact same. If the curves are really that overlapping between 2 independent experiments I would be very surprised. I also can't find experimental details regarding how they purified monocytes, and they only show a reduction in T cells and not NK cells in Fig S3. Another simple experiment would be to test if B2M knockout in cancer cells impairs response? This experiment would help to determine if PBMC T cell alloreactivity is involved, vs NK cell activity, since one would predict loss of killing with B2M KO if there is a role for T cells, or increased killing if NK cells. Or no effect if this is purely related to macrophage activation and direct tumoricidal activity as their model suggests.

- We re-formatted the original figures 3D and 3E to differentiate the PBMC and monocyte co-culture conditions, used larger open symbols to differentiate the monocyte co-cultures curves from PBMCs. We appreciate the Reviewer's point that many immune cell types may be involved, to varying degrees, in the anti-tumor response to STING agonism. As discussed in our response to the previous question, we are not asserting that myeloid cells are the only immune cells involved in the anti-tumor activity of the tumor cell-targeted STING ADCs, but rather that they drive the initial innate immune response, since the ADC delivers STING agonist into cancer cells and FcγR1 expressing myeloid cells. Other cell types, such as NK cells and T cells are expected to be activated in response to the initial type I IFN responses (see also the response to Reviewer #2 – Q2 above). A recent work from David Barbie's lab at DCFI showed that the STING activation in cancer cells stimulates NK cell cancer cell-killing activity in fresh human mesothelioma tumor cultures (Knelson et al. doi: 10.1158/2326-6066). The ability of NK cells to kill B2M KO cancer cells independent from T cells downstream of STING agonism has been shown (Nicolai et al. doi:10.1126/sciimmunol.aaz2738) and we appreciate it's an interesting idea to test this in our model system; however, generation of B2M KO cells will take a long time and the results would not affect the overall conclusion of our manuscript.
- The data we provide in the manuscript suggest that the strong innate immune activation in myeloid cells and cancer cells is sufficient to induce robust cancer cell killing. To complement these results, we added the comparison of cancer cell killing activity induced in cancer cell co-cultures with PBMCs vs monocyte-depleted PBMCs (see new Supplementary Fig. S11A, S11B), in which we used a CD14+ monocyte magnetic depletion method. The cancer cell-killing activity of the tumor cell-targeted ADC is significantly reduced in the monocyte-depleted PBMC co-cultures (as opposed to retaining its activity in the monocyte-enriched PBMCs in original Fig.3D & 3E), strongly suggesting that the FcγR1-expressing myeloid cells are the main cell types that mediate cancer cell-killing activity in PBMC co-cultures. In addition, we also include new data demonstrating that the supernatants (conditioned media) from the cancer cell and PBMC co-cultures are capable of inducing robust killing of the cancer cell monocultures, which indicate

direct tumoricidal activity (see new Supplementary Fig. S11C-S11G). Of course, this does not exclude the involvement of the lymphocyte mediated killing as a secondary mechanism in the tumor microenvironment, which we discuss further in the manuscript as mentioned above.

- We thank the Reviewer for pointing out that we did not describe the monocyte enrichment method; this was an oversight, and we added the method to the revised manuscript (Methods section, Page 36). We used a magnetic enrichment method (Stem Cell Technologies) to isolate CD14+ monocytes with CD16 depletion. The NK cell frequency is quite low in PBMCs (~5-10%), and low number of NK cells in the assay would unlikely be sufficient to induce significant cancer cell killing activity. In the enriched monocytes, due to CD16 depletion, the number of NK cells is even lower. Therefore, in this assay setting, the contribution from NK cells if any, would likely be limited.

Reviewer #3 (Remarks to the Author):

This study aims to test an ADC that targets HER2 or SLC34A2 to deliver STING agonists as a payload. The study team showed that ADC was internalized to myeloid cells in a FcR-I and tumor antigen-dependent fashion. This ADC induces type III interferon, which contributes to an anti-tumor immune response. The main conclusion is that a previously underappreciated role of type III IFN in anti-tumor activity can be elicited by STING-inducing ADC. Although the system appears robust in cytokine production, the study lacks the rigor to generate sufficient impact and conceptual innovation. The study is heavily dependent on in vitro co-culture and SCID mice. STING-mediated immunity depends on CD8 T cells. Syngeneic models and GEMMs are more appropriate to advance this line of agents conceptually. The toxicity study is very limited. Type III IFN shares many signaling components with type I IFN but target mucosal surfaces. It is not surprising that this product induces type III as well, but the significance of type III-specific target tissue is not addressed. Without the use of syngeneic models that utilize IFNLR deficient tumors or IFNLR-deficient hosts, it is not possible to definitely reach the central conclusion that type III contributes to anti-tumor immunity in vivo. In addition, there are specific comments below.

Responses to general comments:

- The study is heavily dependent on in vitro co-culture and SCID mice. STING-mediated immunity depends on CD8 T cells. Syngeneic models and GEMMs are more appropriate to advance this line of agents conceptually.
 - Our studies explored the anti-tumor innate immune activation mediated by the tumor cell-targeted STING agonist ADCs, and we provide substantial evidence for the mechanism of direct anti-tumor activity mediated by these innate immune responses. This mechanism is not mutually exclusive with mechanisms based on CD8 T cells. The mechanism we have unraveled in this study is underappreciated

in the field and, as a T cell-independent mechanism, adds a new layer to the field's understanding of STING-mediated anti-tumor activity and therapeutic approaches.

- The toxicity study is very limited.

- The tumor cell-targeted ADCs used in our in vivo studies did not lead to any clinical observations in mice or any significant systemic cytokine induction; see original main Fig. 2. The ADCs used in these studies are intended for providing proof of concept for tumor cell-targeted STINGa-ADCs, as well as tools for studying their mechanism of action, therefore no further tox studies were performed. We have done extensive preclinical tox studies with our clinical candidate XMT-2056, a HER2-targeted STINGa-ADC bearing a closely related STING agonist, which will be discussed in a separate future manuscript. In the IND-enabling XMT-2056 studies in non-human primates, no adverse events were observed at any dose level.

- Type III IFN shares many signaling components with type I IFN but target mucosal surfaces. It is not surprising that this product induces type III as well, but the significance of type III-specific target tissue is not addressed. Without the use of syngeneic models that utilize IFNLR deficient tumors or IFNLR-deficient hosts, it is not possible to definitely reach the central conclusion that type III contributes to anti-tumor immunity in vivo.

- This comment is conceptually interesting, but unfortunately the proposed studies in mice are not possible. IFNL1, which is the isoform that we found to be a key factor for the ADC anti-tumor activity, is NOT expressed in mice, as discussed in the manuscript; and more generally Type III IFN induction in mouse is relatively weak compared to humans. Thus, it is not possible to study this aspect in mice. For this reason, we focused our Type III IFN studies in human cancer cell and primary human PBMC co-cultures (Fig. 4) and confirmed our findings on Type III IFN induction in fresh primary human tumor cultures *ex vivo* (Fig 7). It is well-known in the field (as we confirmed by speaking with KOLs in the field) that commercially available anti-IFNLR antibodies don't work well; we tested several of those antibodies anyway, and couldn't get them to work consistent with the findings of others in the field. Regardless, IFNLR is reported to be expressed on epithelial cells (including cancer cells) and some immune cell subtypes, especially following innate immune activation (Zanoni et al. doi: [10.3389/fimmu.2017.01661](https://doi.org/10.3389/fimmu.2017.01661)). Thus, our findings suggesting that the type III IFN pathway is involved downstream of STING activation in tumors, perhaps as a first line of defense leading to stronger type I IFN responses akin to the responses seen with pathogen infections in epithelial tissues (see discussion in the manuscript in page 26, 1st paragraph), is not inconsistent with the literature on type III IFN pathway.

1. The heavy use of SCID mice dampens the rigor of the entire study. For example, Figure 2 concludes that ADC is more effective than diABZI. It is known that STING-mediated immunity depends on T cells. Although ADC might be more effective in this setting, a fair comparison would be in immunocompetent mice.

- While the involvement of T cells (adaptive immunity) is desired and expected for immunological memory and durable clinical responses, STING is an innate immune pathway, and the significance of this study is to elucidate the mechanism by which activation of STING leads to direct anti-tumor activity via innate immune pathways. Our results are not inconsistent with or mutually exclusive of the additional contributions of T cells as a secondary mechanism.
- We and others have demonstrated the anti-tumor activity of STING activation in immune-competent models as well as immune-compromised models. We demonstrated efficacy in two immune-competent models based on 4T1 (new Supplementary Fig. S7) and EMT6 (See Figures for responses to Reviewer#1 Q8b); in immune-competent as well as immune-compromised systems, we consistently observed the superior anti-tumor activity of the ADC over free STING agonist. In addition, Wu et al. (reference # 48 in the manuscript) have shown the anti-tumor activity of the tumor cell-targeted STINGa-ADCs in several syngeneic models. Overall, there are many consistent findings of the anti-tumor activity of STING activation across many mouse strains and tumor models, and in studies from different groups.

2. The toxicity studies are quite rudimentary. Mice are more tolerant to immune-related toxicities. No change in body weight does not mean much. In fact, the worrisome clinical toxicity performance (the first case who was treated with ADC passed away) suggests that a more comprehensive toxicity study in large animals, including NHP, is needed. It is possible that the patient died of other reasons. However, the study presented here lack key elements for a clinical-grade new therapeutic agent.

- The focus of this manuscript is the mechanism of action of STING agonist ADCs. As such the manuscript is of high interest to researchers in the STING field as well as researchers who are developing STING agonist-based therapeutics for oncology. This manuscript is not describing a clinical compound. The clinical agent referred to in this comment, XMT-2056, was studied extensively in robust IND-enabling nonclinical toxicology studies, though unfortunately (and not uncommonly in the immunology field) there are significant gaps in translation between nonclinical species and cancer patients. These results will be published separately so that the field may learn from our experience. To be clear, FDA has since lifted the clinical hold in October 2023 on our XMT-2056 Phase I study and we are resuming the trial.

3. Tumor antigen expression has high variability. The ADC effectiveness appears to be dependent on tumor antigen expression. It is unclear how effective this treatment would be for tumors with heterogenous antigen expression levels. The single-antigen design further dampens its clinical potential.

- This question can be generalized to a trade-off with all targeted therapies – whether an ADC, or a tyrosine kinase inhibitor such as EGFRi, targeted therapies

are effective when the tumor expresses and/or depends upon the target – and by definition, less/not effective when the target is absent. This is not a reason to abandon targeted therapies; to the contrary, targeted therapies have revolutionized cancer therapy. FDA-approved ADCs are particularly effective against tumors that express the target – thus, depending on the specifics of the target expression and tumor type, patients are selected based on target expression.

There are 2 important points about the STING-agonist ADCs described here; *i.* Activation of STING in a target-dependent manner results in the killing of target-expressing cells, as well as target-negative cells, on account of the mechanism of action. We demonstrate that the soluble factors released after ADC treatment induce cancer cell death in both target positive or target negative cancer cells via direct tumoricidal activity (new Supplementary Fig. S11C- S11G). Consistent with those findings, we provide additional data for the reviewer only demonstrating the ability of the tumor cell-targeted STINGa ADCs to induce death of the target negative cells (MDA-MB-231-NR cells traced) in the context of co-culture with target-positive cancer cells (SKBR3 WT cells) and PBMCs (figures for Reviewer #3-Q3). This data is generated with our clinical candidate XMT-2056 and is included in a separate manuscript on XMT-2056. These data indicate that the STING-agonist ADCs can be effective against tumors with heterogeneous expression of the target.

Reviewer #3 - Q3:

For reviewer only - will not be included in the manuscript

Flow cytometry analysis of HER2 Expression

Figures demonstrating the immune-mediated bystander killing of HER2 negative cancer cells by XMT-2056. The viability of the MDA-MB-231-NR cells (HER2-negative) in PBMC co-cultures alone or in the presence of SKBR3 WT cells at the indicated ration were traced in an IncuCyte instrument. XMT-2056 induces killing of the HER2 negative MDA-MB-231-NR cells only in the presence of the HER2-expressing SKBR3 cells.

ii. The purpose of the manuscript is to describe the mechanism of action of this class of ADC, not to focus on one particular clinical compound. ADC platforms (such as vcMMAE and Dxd) have been applied to multiple targets, resulting in a suite of targeted therapies from which the treatment can be selected based on the individual tumor.

4. Many cancer cells are defective in STING signaling. The conclusion that cancer cell STING is required for type III IFN production needs many more cell lines than a single cell line.

- We have presented several lines of evidence that the type III response is dependent on cancer cells. In addition to the SKBR3 STING KO cell and immune cell co-cultures, we show that the type III IFN is not induced when the immune cells are cultured on HER2-coated plates (main figures 5B-5E) despite efficient CXCL10 induction. Moreover, the immune cell targeted STINGa-ADC (CD11b-ADC) induces robust CXCL10 in the presence of both STING wt and STING KO cancer cells but induces type III IFN only in the STING wt cancer cell co-cultures. Together these results indicate that the tumor cell-intrinsic STING activity is required for the robust type III IFN production even in the presence of intact immune cell STING activation.

5. Type III IFN and type I IFN share many signaling components. It is not surprising that this particular ADC can induce type I and type III IFNs. A more important question is how therapeutically significant type III IFN is in this setting. There is no attempt to address whether the deficiency in cancer cell specific or host specific type III IFN receptor would make a difference.

- This study is the first to demonstrate that type III IFN is induced in response to STING agonism in cancer cells and in tumors. This finding has important implications for the field in terms of understanding the mechanism of action of STING agonism, and provide a framework for the translational research and biomarker efforts to study tumor intrinsic STING and type III IFN in the clinic.
- We demonstrated Type III IFN induction in cancer cell and immune cell co-cultures *in vitro* (main Fig. 4), in human tumor xenografts *in vivo* (main Fig 4C-4D), and in fresh human tumor fragment cultures *ex vivo* (main Fig 7G). We also demonstrated that neutralizing type III IFN inhibits the cancer cell-killing activity in PBMC co-cultures with two cancer cell lines and with STINGa-ADCs against two different targets (main Fig. 6, Supplementary Fig S17). Our findings were consistent across all of these experimental systems.
- There is no evidence in the literature indicating any deficiencies in cancer cell or host specific type III IFN receptor expression, however we agree that this could be a potential resistance mechanism inhibiting the anti-tumor activity of the STING agonism. Elucidating the potential impact of type III IFN receptor expression

patterns in tumors requires extensive studies that are currently technically challenging and also beyond the scope of this work.

6. Type III IFN has a different target tissue, which is usually mucosa. The contribution of type III signaling-specific target tissue in ADC-mediated anti-tumor immune response is not addressed.

- Yes: it has been reported that in normal physiology in response to viral infections the type III IFN responses take place in mucosal tissues. Type III IFN receptors are expressed on epithelial cells as well as on some immune cell sub populations especially upon infections (see Zannoni et al. doi: [10.3389/fimmu.2017.01661](https://doi.org/10.3389/fimmu.2017.01661) for review of the field); thus the literature supports the concept that the receptor may be expressed on epithelial cancer cells and immune cells subtypes.

7. Cancer cells are defective for STING signaling for many reasons. The exclusive focus on STING expression level is narrow in scope and not novel.

- Please see the comment by the reviewer #2 (first comment) and our response. STING silencing in tumors has been shown, however there is a misconception in the field about the frequency of the STING silencing in cancer cells, which is largely based on cancer cell monoculture studies; in this study we have unraveled why those monoculture studies are prone to significant artifact (see Results section “Cues from primary human immune cells potentiate STING pathway activation in cancer cell monocultures” Supplementary Fig. S13). Our finding that cancer cell STING expression/activation can increase in response to cues from immune cells and the TME (Supplementary Fig. S13) is a novel observation (as noted by the Reviewers 1 and 2) as well as by experts in the STING field that we have engaged during this research.

8. The impact of ADC on T cells in the TME and draining lymph nodes are largely ignored in this study.

- The focus of this manuscript is the innate immune responses to the STING-ADCs and in particular the activation of STING in both myeloid cells and cancer cells. These findings are not exclusive of the role of T cells in longer term anti-tumor immune response. We have added to the Discussion to explicitly discuss this point (page 25, last paragraph continued on page 26).

REVIEWERS' COMMENTS

Reviewer #1 (Remarks to the Author):

Dear editor,

This manuscript by Cetinbas et al, after careful revision work, has been largely improved by the authors. Overall, this is considerable work describing the therapeutic potential of a new class of antibody-drug conjugates that will undoubtedly be of interest to the oncology community. A multitude of different aspects of the mechanism of action as well as safety concerns related to these new compounds were generally well defined by the authors in the revised manuscript through additional work, making this paper worthy of publication in its current form.

Reviewer #2 (Remarks to the Author):

The authors have done a nice job and satisfactorily addressed my concerns.

Reviewer #4 (Remarks to the Author):

The authors have developed an overall thoughtful response. Although mice do not express L1, they do express L2 and L3. Little evidence is provided to show L2/3 is not important for the phenotype. Given this is a translation-targeted product, additional toxicity studies in large animal models are essential. Head-to-head comparison with other STING assets needs to be done in immunocompetent mouse models. GEMMs need to be utilized. Cancer cell-specific type III IFN activation remains to be substantiated. Although conceptually interesting, the experiments lack the essential rigor to justify the main conclusions. The response to some of the questions raised by the other two reviewers appears to be marginal as well.

Response to 2nd Round Reviewer Comments -- Cetinbas *et al.*

Reviewer #1 (Remarks to the Author):

This manuscript by Cetinbas *et al.*, after careful revision work, has been largely improved by the authors. Overall, this is considerable work describing the therapeutic potential of a new class of antibody-drug conjugates that will undoubtedly be of interest to the oncology community. A multitude of different aspects of the mechanism of action as well as safety concerns related to these new compounds were generally well defined by the authors in the revised manuscript through additional work, making this paper worthy of publication in its current form.

The authors are grateful to the Reviewer for their time and effort to review our revised manuscript.

Reviewer #2 (Remarks to the Author):

The authors have done a nice job and satisfactorily addressed my concerns.

The authors are grateful to the Reviewer for their time and effort to review our revised manuscript.

Reviewer #4 (Remarks to the Author):

The authors have developed an overall thoughtful response.

Although mice do not express L1, they do express L2 and L3. Little evidence is provided to show L2/3 is not important for the phenotype.

The authors are grateful to the Reviewer for their time and effort to review our revised manuscript.

We tested an anti-L2/L3 antibody and found that it did not have an effect in the assay, whereas the anti-L1 antibody did. The revised manuscript addressed the question of L1 vs. L2 and L3 in both the Results and the Discussion sections (excerpts below). We concluded that L1 “may be the key factor” which is consistent with our data. Further studies beyond the scope of this manuscript will be informative, and we added to the Discussion to explicitly state the limitation of mouse studies for this purpose:

Added to Discussion: “Therefore, future studies addressing the interplay between cancer-cell STING signaling and type III IFN production in the TME, as well as how *IFNλ1* mediates STING-induced anti-tumor activity, could be highly insightful. Of

note, the utility of mouse studies for this purpose will be limited since mouse does not express IFNλ1.”

From Results section in reviewed manuscript: “Indeed, IFNλ1-neutralizing antibodies countered the cancer cell killing activity induced by HER2 ADC, as evidenced by a ~6-fold increase in the EC₅₀ (Fig. 6A). IFNλ2-neutralizing antibodies had no effect in this assay. Of note, anti-IFNλ1 antibody is cross-reactive with IFNλ2 and IFNλ3, while the anti-IFNλ2 antibody is cross-reactive with IFNλ3. Therefore, these data suggest that IFNλ1 may be the key type III IFN mediating the observed anti-tumor activity downstream of STING agonism.”

From Discussion in reviewed manuscript: “Our data indicate that type III IFNs, specifically IFNλ1, are required for type I IFN production and the anti-tumor activity elicited by cancer-cell-targeted STINGa ADCs.”

Given this is a translation-targeted product, additional toxicity studies in large animal models are essential. Head-to-head comparison with other STING assets needs to be done in immunocompetent mouse models. GEMMs need to be utilized. Cancer cell-specific type III IFN activation remains to be substantiated. Although conceptually interesting, the experiments lack the essential rigor to justify the main conclusions. The response to some of the questions raised by the other two reviewers appears to be marginal as well.

We agree that many future studies could continue to inform the pursuit of modulating STING signaling to realize anti-tumor activity. We added a sentence to the Discussion that mentions some of the Reviewer’s suggestions:

Added to Discussion: “It will be informative to compare STINGa ADCs to free STINGa in immunocompetent mouse models including genetically engineered mouse models (GEMMs) and ultimately in clinical trials.”

This manuscript describes the mechanism-of-action for STINGa ADCs; it is not describing a particular clinical compound or a translation-targeted product. None of the ADCs in this manuscript are intended for clinical development, which is why we cannot evaluate them in large animal toxicity studies.